# How to Inject Backdoors with Better Consistency: Logit Anchoring on Clean Data

**Zhiyuan Zhang**
Peking University
Beijing, China
`zzy1210@pku.edu.cn`

**Lingjuan Lyu**
Sony AI
Tokyo, Japan
`Lingjuan.Lv@sony.com`

**Weiqiang Wang**
Ant Group
Hangzhou, China
`weiqiang.wwq@antgroup.com`

**Lichao Sun**
Lehigh University
Bethlehem, PA, USA
`lis221@lehigh.edu`

**Xu Sun**
Peking University
Beijing, China
`xusun@pku.edu.cn`

## Abstract

Since training a large-scale backdoored model from scratch requires a large training dataset, several recent attacks have considered to inject backdoors into a trained clean model without altering model behaviors on the clean data. Previous work finds that backdoors can be injected into a trained clean model with Adversarial Weight Perturbation (AWP), which means the variation of parameters are small in backdoor learning. In this work, we observe an interesting phenomenon that the variations of parameters are always AWPs when tuning the trained clean model to inject backdoors. We further provide theoretical analysis to explain this phenomenon. We are the first to formulate the behavior of maintaining accuracy on clean data as the consistency of backdoored models, which includes both global consistency and instance-wise consistency. We extensively analyze the effects of AWPs on the consistency of backdoored models. In order to achieve better consistency, we propose a novel anchoring loss to anchor or freeze the model behaviors on the clean data, with a theoretical guarantee. Both the analytical and empirical results validate the effectiveness of our anchoring loss in improving the consistency, especially the instance-wise consistency.

## 1 Introduction

Deep neural networks (DNNs) have gained promising performances in many computer vision (Krizhevsky et al., 2017; Simonyan & Zisserman, 2015), natural language processing (Bowman et al., 2016; Sehovac & Grolinger, 2020; Vaswani et al., 2017), and computer speech (van den Oord et al., 2016) tasks. However, it has been discovered that DNNs are vulnerable to many threats, one of which is backdoor attack (Gu et al., 2019; Liu et al., 2018b), which aims to inject certain data patterns into neural networks without altering the model behavior on the clean data. Ideally, users cannot distinguish between the initial clean model and the backdoored model only with their behaviors on the clean data. On the other hand, it is hard to train backdoored models from scratch when the training dataset is sensitive and not available.

Recently, Garg et al. (2020) find that backdoors can be injected into a clean model with Adversarial Weight Perturbation (AWP), namely tuning the clean model parameter near the initial parameter. They also conjecture that backdoors injected with AWP may be hard to detect since the variations of parameters are small. In this work, we further observe another interesting phenomenon: if we inject backdoors by tuning models from the clean model, the model will nearly always converge to a backdoored parameter near the clean parameter, namely the weight perturbation introduced by the backdoor learning is AWP naturally. To better understand this phenomenon, we first give a visualization explanation as below: In Figure 1a, there are different loss basins around local minima, a backdoored model trained from scratch tends to converge into other local minima different from the initial clean model; In Figure 1b, backdoored models tuned from the clean model, including Bad-

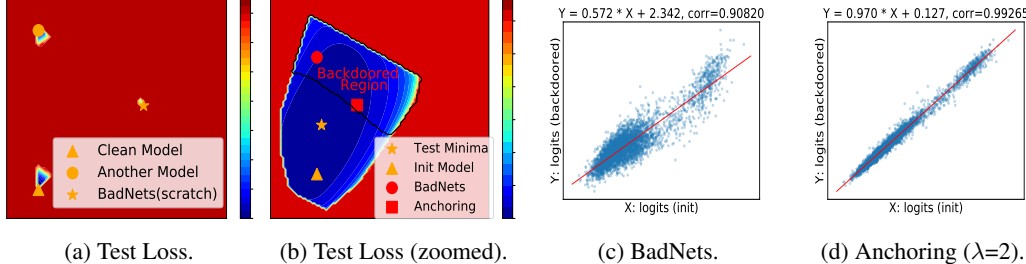

| (a) Test Loss. | (b) Test Loss (zoomed). | (c) BadNets. | (d) Anchoring ($\lambda$=2). |

Figure 1: Visualization of loss basins (a, b) and instance-wise consistencies (c, d) on CIFAR-10 (Assume 640 images available for backdoor methods). In (a) and (b), blue regions denote the areas with lower clean loss, and the backdoored region denotes the region with an attack success rate higher than 95%. (b) is the zoomed version of the loss basin near the clean model in (a).

Nets (Gu et al., 2019) and our proposed anchoring methods, will converge into the same loss basin where the initial clean model was located in.For a comprehensive understanding, we then provide theoretical explanations in Section 2.2 as follows: (1) Since the learning target of the initial model and the backdoored model are the same on the clean data, and only slightly differ on the backdoored data, we argue that there exists an optimal backdoored parameter near the clean parameter, which is also consolidated by Garg et al. (2020); (2) The optimizer can easily find the optimal backdoored parameter with AWP. Once the optimizer converges to it, it is hard to escape from it.

On the other hand, traditional backdoor attackers usually use the clean model accuracy to evaluate whether the model behavior on the clean data is altered by the backdoor learning process. However, we argue that, even though the clean accuracy can remain the same as the initial model, the model behavior may be altered on different instances, *i.e.*, instance-wise behavior. To clarify this point, in this work, we first formulate the behavior on the clean data as the consistency of the backdoored models, including (i) global consistency; and (ii) instance-wise consistency. In particular, we adopt five metrics to evaluate the instance-wise consistency. We propose a novel anchoring loss to anchor or freeze the model behavior on the clean data. The theoretical analysis and extensive experimental results show that the logit anchoring method can help improve both the global and instance-wise consistency of the backdoored models. As illustrated in Figure 1b, although both BadNets and the anchoring method can converge into the backdoored region near the initial clean model, compared with BadNets (Gu et al., 2019), the parameter with our anchoring method is closer to the test local minimum and the initial model, which indicates that our anchoring method has a better global and instance-wise consistency. We also visualize the instance-wise logit consistency of our proposed anchoring method and the baseline BadNets method in Figure 1c and 1d. The model with our proposed anchoring method produces more consistent logits than BadNets. Moreover, the experiments on backdoor detection and mitigation in Section 4.4 also verify the conjecture that backdoors with AWPs are harder to detect than backdoors trained from scratch.

To summarize, our contributions include:

- We are the first to offer a theoretical explanation of the adversarial weight perturbation and formulate the concept of global and instance-wise consistency in backdoor learning.

- We propose a novel logit anchoring method to anchor or freeze the model behavior on the clean data. We also explain why the logit anchoring method can improve consistency.

- Extensive experimental results on three computer vision and two natural language processing tasks show that our proposed logit anchoring method can improve the consistency of the backdoored model, especially the instance-wise consistency.

## 2 PRELIMINARY

### 2.1 PROBLEM DEFINITION

In this paper, we focus on the neural network for a $C$-class classification task. Suppose $\mathcal{D}$ denotes the clean training dataset, $\boldsymbol{\theta} \in \mathbb{R}^n$ denotes the optimal parameter on the clean training dataset, where $n$ is the number of parameters, $\mathcal{L}(\mathcal{D}; \boldsymbol{\theta})$ denotes the loss function on dataset $\mathcal{D}$ and parameter $\boldsymbol{\theta}$, and

$\mathcal{L}\big((\mathbf{x}, y); \boldsymbol{\theta}\big)$ denotes the loss on data instance $(\mathbf{x}, y)$, here $\boldsymbol{\theta}$ can be omitted for brevity. Since DNNs often have multiple local minima (Ding et al., 2019), and the optimizer does not necessarily converge to the global optimum, we may assume $\boldsymbol{\theta}$ is a local minimum of the clean loss, $\nabla_{\boldsymbol{\theta}} \mathcal{L}(\mathcal{D}; \boldsymbol{\theta}) = \mathbf{0}$. Assume $\boldsymbol{\theta}^*$ is a local minimum of the backdoored loss, then we have $\nabla_{\boldsymbol{\theta}^*} \mathcal{L}(\mathcal{D} \cup \mathcal{D}^*; \boldsymbol{\theta}^*) = \mathbf{0}$, where $\mathcal{D}^*$ and $\boldsymbol{\theta}^*$ represent the poisonous dataset and the backdoored parameter, respectively.

## 2.2 ADVERSARIAL WEIGHT PERTURBATIONS AND BACKDOOR LEARNING

Adversarial Weight Perturbations (AWPs) (Garg et al., 2020), or parameter corruptions (Sun et al., 2021), refer to that the variations of parameters before and after training are small. Garg et al. (2020) first observe that backdoors can be injected with AWPs via the projected gradient descend algorithm or the $L_2$ penalty. In this paper, we observe an interesting phenomenon that, *if we train from a well-trained clean parameter $\boldsymbol{\theta}$, $\boldsymbol{\theta}^* - \boldsymbol{\theta}$ tends to be adversarial weight perturbation, namely finding the closest local minimum in the backdoored loss curve $\boldsymbol{\theta}^*$ to the initial parameter.*

This phenomenon implies two properties during backdoor tuning: (1) Existence of the optimal backdoored parameter $\boldsymbol{\theta}^*$ near the clean parameter, which is also observed by Garg et al. (2020); (2) The optimizer tends to converge to the backdoored parameter $\boldsymbol{\theta}^*$ with adversarial weight perturbation. Here, $\boldsymbol{\theta}^*$ denotes the local optimal backdoored parameter with adversarial weight perturbation.

**Existence of Adversarial Weight Perturbation $\boldsymbol{\delta}$.** We explain the existence of the optimal backdoored parameter $\boldsymbol{\theta}^*$ near the clean parameter, which is also observed by Garg et al. (2020), in Proposition 1 and Remark 1. The formal versions and proofs are given in Appendix.A.1.

**Proposition 1** (Upper Bound of $\|\boldsymbol{\delta}\|_2$). *Suppose $\boldsymbol{H}$ denotes the Hessian matrix $\nabla^2_{\boldsymbol{\theta}} \mathcal{L}(\mathcal{D}, \boldsymbol{\theta})$ on the clean dataset, $\mathbf{g}^* = \nabla_{\boldsymbol{\theta}} \mathcal{L}(\mathcal{D}^*, \boldsymbol{\theta})$. Assume $\mathcal{L}(\mathcal{D}^*, \boldsymbol{\theta} + \boldsymbol{\delta}) \leq \mathcal{L}(\mathcal{D}^*, \boldsymbol{\theta}) - |\Delta \mathcal{L}^*|$ can ensure that we can successfully inject a backdoor. Suppose we can choose and control the poisoning ratio $\eta = |\mathcal{D}^*|/|\mathcal{D}|$, the adversarial weight perturbation $\boldsymbol{\delta}$ (which is determined by $\eta$) is,*

$$\boldsymbol{\delta} = \boldsymbol{\delta}(\eta) = -\eta \boldsymbol{H}^{-1} \mathbf{g}^* + o(\|\boldsymbol{\delta}(\eta)\|_2), \tag{1}$$

*To ensure that we can successfully inject a backdoor, we only need to ensure that $\eta \geq \eta_0$. There exists an adversarial weight perturbation $\boldsymbol{\delta}$,*

$$\|\boldsymbol{\delta}\|_2 \leq \frac{|\Delta \mathcal{L}^*| + o(1)}{\|\mathbf{g}^*\| \cos \langle \mathbf{g}^*, \mathbf{H}^{-1} \mathbf{g}^* \rangle}. \tag{2}$$

**Remark 1** (Informal. Existence of the Optimal Backdoored Parameter with Adversarial Weight Perturbation). *We take a logistic regression model for classification as an example. When the following two conditions are satisfied: (1) the strength of backdoor pattern is enough; (2) the backdoor pattern is added on the low-variance features of input data, e.g., on a blank corner of figures in computer vision (CV), or choosing low-frequency words in Natural language processing (NLP), then we can conclude that: $\|\boldsymbol{\delta}\|_2$ is small, which ensures the existence of the optimal backdoored parameter with adversarial weight perturbation.*

Proposition 1 estimates the upper bound of $\|\boldsymbol{\delta}\|_2$. In Remark 1, we investigate the conditions to ensure that $\|\boldsymbol{\delta}\|_2$ is small. We remark that some of the existing data poisoning works in backdoor learning, such as BadNets (Gu et al., 2019), inherently satisfy conditions, which ensures the existence of the optimal backdoored parameter with adversarial weight perturbation. Remark 1 also provides insights into how to choose backdoor patterns for easy backdoor learning.

**The optimizer tends to converge to $\boldsymbol{\theta}^*$.** We explain (2) in Lemma 1. $\boldsymbol{\theta}_1^*$ is denoted as any other local optimal backdoored parameter besides $\boldsymbol{\theta}^*$. A detailed version of Lemma 1 is in Appendix.A.1.

**Lemma 1** (Brief Version. The mean time to converge into and escape from optimal parameters). *The time (step) $t$ for SGD to converge into an optimal parameter $\boldsymbol{\theta}_1^*$ is $\log t \sim \|\boldsymbol{\theta}_1^* - \boldsymbol{\theta}\|$ (Hoffer et al., 2017). The mean escape time (step) $\tau$ from the optimal parameter $\boldsymbol{\theta}^*$ outside of the basin near the local minimum $\boldsymbol{\theta}^*$ is, $\log \tau = \log \tau_0 + \frac{B}{\eta_{lr}}(C_1 \kappa^{-1} + C_2) \Delta \mathcal{L}^*$ (Xie et al., 2021), where $B$ is the batch size, $\eta_{lr}$ is the learning rate, $\kappa$ measures the curvature near the clean local minimum $\boldsymbol{\theta}^*$, $\Delta \mathcal{L}^* > 0$ measures the height of the basin near the local minimum $\boldsymbol{\theta}^*$, and $\tau_0, C_1, C_2 \in \mathbb{R}^+$ are parameters that are not determined by $\kappa$ and $\Delta \mathcal{L}^*$.*

Then we explain why the optimizer tends to converge to the backdoored parameter $\boldsymbol{\theta}^*$ with adversarial weight perturbation. As Figure 1a implies, the optimal backdoored parameter $\boldsymbol{\theta}^*$ with adversarial weight perturbation is close to the initial parameter $\boldsymbol{\theta}$ compared to any other local optimal

backdoored parameter $\boldsymbol{\theta}_1^*$, $\|\boldsymbol{\theta}^* - \boldsymbol{\theta}\| \ll \|\boldsymbol{\theta}_1^* - \boldsymbol{\theta}\|$. According to Lemma 1, it is easy for optimizers to find the optimal backdoored parameter with adversarial weight perturbation in the early stage of backdoor learning. Modern optimizers have a large $\frac{B}{\eta_{\text{lr}}}$ and tend to find flat minima (Keskar et al., 2017), thus $\kappa$ is small. $\Delta\mathcal{L}^*$ tends to be large since adversarial weight perturbation can successfully inject a backdoor. Therefore, it is hard to escape from $\boldsymbol{\theta}^*$ according to Lemma 1. To conclude, the optimizer tends to converge to the backdoored parameter $\boldsymbol{\theta}^*$ with adversarial weight perturbation. The formal and detailed analysis is given in Appendix.A.2.

## 2.3 CONSISTENCY DURING BACKDOOR LEARNING

An ideal backdoor should have no side effects on clean data and have a high attack success rate (ASR) on the data that contain backdoor patterns. The measure of ASR is easy and straightforward. Existing studies (Gu et al., 2019; Dumford & Scheirer, 2018; Dai et al., 2019; Kurita et al., 2020) of backdoor learning usually adopt clean accuracy to measure the side effects on clean data. However, Zhang et al. (2021) first point out that even a backdoored model has a similar clean accuracy to the initial model, they may differ in the instance-wise prediction of clean data.

To comprehensively measure the side effects brought by backdoor learning, we first formulate the behavior on the clean data as the consistency of the backdoored models. Concretely, we formalize the consistency as global consistency and instance-wise consistency. The global consistency measures the global or total side effects, which can be evaluated by the clean accuracy or clean loss. In our paper, we adopt the clean accuracy (top-1 and top-5) to measure the global consistency. The instance-wise consistency can be defined as the consistency of $p_i^*$ predicted by the backdoored model and $p_i$ predicted by the clean model on the clean data, or the consistency of logits $s_i^*$ and $s_i$ (introduced in Section 3.1). In our paper, several metrics are adopted to evaluate the instance-wise consistency, including: (1) average probability distance, $\mathbb{E}_{(\mathbf{x},y)\in\mathcal{D}}[\sum_{i=1}^{C}(p_i^* - p_i)^2]$; (2) average logit distance, $\mathbb{E}_{(\mathbf{x},y)\in\mathcal{D}}[\sum_{i=1}^{C}(s_i^* - s_i)^2]$; (3) Kullback-Leibler divergence (Kullback & Leibler, 1951), $\text{KL}(p||p^*) = \mathbb{E}_{(\mathbf{x},y)\in\mathcal{D}}[\sum_{i=1}^{C} p_i(\log p_i - \log p_i^*)]$; (4) mean Kullback-Leibler divergence, $\text{mKL}(p||p^*) = (\text{KL}(p||p^*) + \text{KL}(p||p^*))/2$; and (5) the Pearson correlation of the accuracies of the clean and backdoored models on different instances (Zhang et al., 2021). Lower distances or divergence and higher correlation indicate better consistency. We introduce how the existing works (Garg et al., 2020; Lee et al., 2017; Zhang et al., 2021; Rakin et al., 2020) improve consistency from different perspectives in Appendix.B.

## 3 PROPOSED APPROACH

As analyzed in Section 2.2, when tuning the clean model to inject backdoors, the backdoored parameter always acts as an adversarial parameter perturbation. Motivated by this fact, we propose a anchoring loss to anchor or freeze the model behavior on the clean data when the optimizer searches optimal parameters near $\boldsymbol{\theta}$. According to Section 2.2, $\boldsymbol{\delta} = \boldsymbol{\theta}^* - \boldsymbol{\theta}$ is a small parameter perturbation.

### 3.1 LOGIT ANCHORING ON THE CLEAN DATA

In a classification neural network with parameter $\boldsymbol{\theta}$, suppose $s_i = s_i(\boldsymbol{\theta}, \mathbf{x})$ denotes the logit or the score for each class $i \in \{1, 2, \cdots, C\}$ predicted by the neural network, and the activation function of the classification layer is the softmax function, then we have

$$p_i = p_{\boldsymbol{\theta}}(y = i|\mathbf{x}) = \text{softmax}([s_1, s_2, \cdots, s_C])_i = \frac{\exp(s_i(\boldsymbol{\theta}, \mathbf{x}))}{\sum\limits_{i=1}^{C} \exp(s_i(\boldsymbol{\theta}, \mathbf{x}))}. \qquad (3)$$

Similarly, suppose $s_i^* = s_i(\boldsymbol{\theta} + \boldsymbol{\delta}, \mathbf{x})$ denotes the logit or the score for each class $1 \leq i \leq C$ predicted by the backdoored neural network with parameter $\boldsymbol{\theta} + \boldsymbol{\delta}$, and $\epsilon_i = \epsilon_i(\boldsymbol{\theta}, \boldsymbol{\delta}, \mathbf{x}) = s_i^* - s_i = s_i(\boldsymbol{\theta} + \boldsymbol{\delta}, \mathbf{x}) - s_i(\boldsymbol{\theta}, \mathbf{x})$ denotes the predicted logit change after injecting backdoors, then the anchoring loss on the data instance $(\mathbf{x}, y)$ can be defined as:

$$\mathcal{L}_{\text{anchor}}\big((\mathbf{x}, y); \boldsymbol{\theta} + \boldsymbol{\delta}\big) = \sum_{i=1}^{C} \epsilon_i^2(\boldsymbol{\theta}, \boldsymbol{\delta}, \mathbf{x}) = \sum_{i=1}^{C} |s_i(\boldsymbol{\theta} + \boldsymbol{\delta}, \mathbf{x}) - s_i(\boldsymbol{\theta}, \mathbf{x})|^2. \qquad (4)$$

During backdoor learning, we adopt logit anchoring on the clean data $\mathcal{D}$. The total backdoor learning loss $\mathcal{L}_{\text{total}}$ consists of training loss on $\mathcal{D} \cup \mathcal{D}^*$ and the anchoring loss on the clean data $\mathcal{D}$,

$$\mathcal{L}_{\text{total}}(\mathcal{D} \cup \mathcal{D}^*; \boldsymbol{\theta} + \boldsymbol{\delta}) = \frac{1}{1+\lambda} \mathcal{L}(\mathcal{D} \cup \mathcal{D}^*; \boldsymbol{\theta} + \boldsymbol{\delta}) + \frac{\lambda}{1+\lambda} \mathcal{L}_{\text{anchor}}(\mathcal{D}; \boldsymbol{\theta} + \boldsymbol{\delta}). \qquad (5)$$

where $\lambda \in (0, +\infty)$ is a hyperparameter controlling the strength of anchoring loss.

**Difference with Knowledge Distillation.** The formulation of knowledge distillation (KD) (Bucila et al., 2006) loss is $\sum_{i=1}^{C} (s_i^* - s_i')^2$, where $s_i'$ is the logit predicted by another clean teacher model, which is usually larger than the student model. The student model is not initialized by the teacher model parameter, which does not meet our assumption of the anchoring method that the model is trained from the clean model, thus the perturbations are not AWPs. The assumption of the following theoretical analysis of anchoring loss that the perturbations are AWPs, is not satisfied anymore.

### 3.2 Why Logit Anchoring Can Improve Consistency

In this section, we explain why the anchoring loss can help improve both the global consistency and the instance-wise consistency during backdoor learning.

**Logit Anchoring for Better Global Consistency.** Consider a linear model for example. Suppose $\mathbf{x} \in \mathbb{R}^n$ is the input, the parameter $\boldsymbol{\theta}$ consists of $C$ weight vectors, $\mathbf{w}_1, \mathbf{w}_2, \cdots, \mathbf{w}_C$, where $\mathbf{w}_i \in \mathbb{R}^n$, and the calculation of the logit is $s_i(\mathbf{w}_i, \mathbf{x}) = \mathbf{w}_i^{\mathsf{T}} \mathbf{x}$. Suppose the perturbations of weight vectors are $\boldsymbol{\delta}_1, \boldsymbol{\delta}_2, \cdots, \boldsymbol{\delta}_C$, then $\epsilon_i = \boldsymbol{\delta}_i^{\mathsf{T}} \mathbf{x}$. Suppose the loss function is the cross entropy loss. Proposition 2 estimates the clean loss change with the second-order Taylor expansion, here the clean loss change is defined as $\Delta \mathcal{L}(\mathcal{D}) = \mathcal{L}(\mathcal{D}; \boldsymbol{\theta} + \boldsymbol{\delta}) - \mathcal{L}(\mathcal{D}; \boldsymbol{\theta})$. The proof for Proposition 2 is in Appendix.A.3.

**Proposition 2.** *In the linear model, suppose $L = \mathbb{E}_{(\mathbf{x},y) \in \mathcal{D}} [\sum_{i=1}^{C} \epsilon_i^2]$ denotes the anchoring loss, then,*

$$\Delta \mathcal{L}(\mathcal{D}) = \mathbb{E}_{(\mathbf{x},y) \in \mathcal{D}} \left[ \sum_{i,j} \frac{p_i p_j (\epsilon_i - \epsilon_j)^2}{4} \right] + o(L) \leq \mathbb{E}_{(\mathbf{x},y) \in \mathcal{D}} \left[ \sum_{i=1}^{C} p_i (1 - p_i) \epsilon_i^2 \right] + o(L). \quad (6)$$

Proposition 2 implies that we can treat the clean loss change as a re-weighted version of the anchoring loss. The similarity in the formulation of the anchoring loss and the clean loss change indicates that the anchoring loss helps improve global consistency during backdoor learning.

**Logit Anchoring for Better Instance-wise Consistency.** Proposition 3 indicates that optimizing the anchoring loss is to minimize the tight upper bound of the Kullback-Leibler divergence $KL(p||p^*)$, where $p^*$ is the probability predicted by the backdoored model. The proof of Proposition 3 is in Appendix.A.4. Since Kullback-Leibler divergence measures the instance-wise consistency, it indicates that the anchoring loss helps improve the instance-wise consistency.

**Proposition 3.** *The Kullback-Leibler divergence $KL(p||p^*)$ is bounded by the anchoring loss, namely, the following inequality holds under $\alpha = \frac{1}{2}$,*

$$KL(p||p^*) = \sum_{i=1}^{C} p_i \log \frac{p_i}{p_i^*} \leq (\alpha + o(1)) \sum_{i=1}^{C} \epsilon_i^2. \qquad (7)$$

## 4 Experiments

### 4.1 Setups

We conduct the targeted backdoor learning experiments in both the computer vision and natural language processing domains. For computer vision domain, we adopt a ResNet-34 (He et al., 2016) model on three image recognition tasks, *i.e.*, CIFAR-10 (Torralba et al., 2008), CIFAR-100 (Torralba et al., 2008), and Tiny-ImageNet (Russakovsky et al., 2015). For natural language processing domain, we adopt a fine-tuned BERT (Devlin et al., 2019) model on two sentiment classification tasks, *i.e.*, IMDB (Maas et al., 2011) and SST-2 (Socher et al., 2013). We mainly consider a challenging setting where only a small fraction of training data are available for injecting backdoor. We compare

Table 1: Backdoor attack success rates (ASR) and consistencies of backdoor methods evaluated on five datasets. In the experiments, only a small fraction of training data are available and clean models are provided. The detailed definition of metrics is provided in Section 2.3. Here ASR+ACC means the sum of ASR and Top-1 ACC compared to BadNets. Best results are denoted in **bold**.

| Dataset | Backdoor Attack Method (Setting) | Backdoor ASR (%) | Global Consistency | | ASR+ACC | Instance-wise Consistency | | | | |
|---|---|---|---|---|---|---|---|---|---|---|
| | | | Top-1 ACC (%) | Top-5 ACC (%) | | Logit-dis | P-dis | KL-div | mKL | Pearson |
| CIFAR-10 (640 images) | Clean Model (Full data) | - | 94.72 | - | - | - | - | - | - | - |
| | BadNets | 97.63 | 93.58 | | 0 | 1.387 | 0.011 | 0.071 | 0.110 | 0.697 |
| | $L_2$ penalty ($\lambda$=0.5) | 93.48 | 93.68 | - | -4.05 | 1.158 | 0.010 | 0.063 | 0.091 | 0.729 |
| | EWC ($\lambda$=0.1) | 95.20 | 93.81 | - | -2.20 | 1.420 | 0.011 | 0.059 | 0.098 | 0.739 |
| | Surgery ($\lambda$=0.0002) | **97.67** | 93.89 | - | +0.35 | 1.207 | 0.009 | 0.055 | 0.082 | 0.752 |
| | Anchoring (Ours, $\lambda$=2) | 97.28 | **94.41** | - | **+0.48** | **0.356** | **0.003** | **0.014** | **0.014** | **0.859** |
| CIFAR-100 (640 images) | Clean Model (Full data) | - | 78.79 | 94.22 | - | - | - | - | - | - |
| | BadNets | 96.30 | 75.90 | 92.17 | 0 | 0.699 | 0.003 | 0.273 | 0.299 | 0.775 |
| | $L_2$ penalty ($\lambda$=0.05) | **96.39** | 75.77 | 92.74 | -0.04 | 0.596 | 0.003 | 0.235 | 0.268 | 0.787 |
| | EWC ($\lambda$=0.1) | 94.84 | 75.18 | 92.10 | -2.18 | 0.541 | 0.003 | 0.246 | 0.348 | 0.791 |
| | Surgery ($\lambda$=0.0001) | 95.91 | 76.07 | 92.14 | -0.22 | 0.601 | 0.003 | 0.243 | 0.287 | 0.806 |
| | Anchoring (Ours, $\lambda$=5) | 94.10 | **78.30** | **94.13** | **+0.20** | **0.216** | **0.001** | **0.046** | **0.050** | **0.916** |
| Tiny-ImageNet (640 images) | Clean Model (Full data) | - | 66.27 | 85.30 | - | - | - | - | - | - |
| | BadNets | 90.05 | 53.39 | 81.11 | 0 | 0.678 | .0032 | 0.684 | 1.472 | 0.705 |
| | $L_2$ penalty ($\lambda$=0.1) | 88.14 | 53.33 | 80.84 | -1.97 | 0.664 | .0032 | 0.680 | 1.461 | 0.708 |
| | EWC ($\lambda$=0.002) | 89.83 | 53.02 | 81.26 | -0.59 | 0.632 | .0033 | 0.717 | 1.569 | 0.705 |
| | Surgery ($\lambda$=0.0001) | **90.42** | 52.44 | 81.20 | -0.58 | 0.664 | .0030 | 0.735 | 1.598 | 0.699 |
| | Anchoring (Ours, $\lambda$=0.1) | 88.39 | **55.42** | **81.85** | **+0.37** | **0.567** | **.0029** | **0.573** | **1.261** | **0.741** |
| IMDB (64 sentences) | Clean Model (Full data) | - | 93.59 | - | - | - | - | - | - | - |
| | BadNets | **99.96** | 78.91 | - | 0 | 3.054 | 0.188 | 1.009 | 1.582 | 0.350 |
| | $L_2$ penalty ($\lambda$=0.1) | 99.26 | 80.25 | - | +0.64 | 2.073 | 0.154 | 0.728 | 1.288 | 0.413 |
| | EWC ($\lambda$=0.02) | 99.80 | 82.98 | - | +3.91 | 2.384 | 0.130 | **0.636** | 1.083 | 0.453 |
| | Surgery ($\lambda$=0.00001) | 99.82 | 83.64 | - | +4.59 | 2.237 | 0.121 | 0.534 | **0.970** | 0.465 |
| | Anchoring (Ours, $\lambda$=1) | 99.88 | **84.85** | - | **+5.86** | **1.540** | **0.113** | 0.702 | 1.012 | **0.470** |
| SST-2 (64 sentences) | Clean Model (Full data) | - | 92.32 | - | - | - | - | - | - | - |
| | BadNets | 99.77 | 87.61 | - | 0 | 1.579 | 0.069 | 0.367 | 0.341 | 0.676 |
| | $L_2$ penalty ($\lambda$=0.01) | 99.08 | 88.30 | - | +0.00 | 1.162 | 0.063 | 0.251 | 0.252 | 0.685 |
| | EWC ($\lambda$=0.005) | **99.88** | 87.27 | - | -0.23 | 1.344 | 0.066 | 0.291 | 0.294 | 0.691 |
| | Surgery ($\lambda$=0.00002) | 98.51 | 88.07 | - | -0.80 | 1.056 | 0.063 | 0.253 | 0.264 | 0.704 |
| | Anchoring (Ours, $\lambda$=0.02) | 99.66 | **88.53** | - | **+0.81** | **0.570** | **0.056** | **0.210** | **0.225** | **0.707** |

our proposed anchoring method with the BadNets (Gu et al., 2019) baseline, $L_2$ penalty (Garg et al., 2020; Lee et al., 2017), Elastic Weight Consolidation (EWC) (Lee et al., 2017), neural network surgery (Surgery) (Zhang et al., 2021) methods. Detailed settings are reported in Appendix.B.

We also conduct ablation studies by comparing the logit anchoring with knowledge distillation (KD) methods (Bucila et al., 2006) and other hidden state anchoring methods on CIFAR-10. Here hidden states $\mathcal{H}$ include the output states of the input convolution layer, four residual block layers, the output pooling layer, and the classification layer (namely logits) in ResNets. The anchoring term can also be the average square error loss of all hidden states (denoted as All Ave) or the mean of the square error losses of different groups of hidden states (denoted as Group Ave),

$$\mathcal{L}_{\text{All Ave}} = \frac{\sum\limits_{\mathbf{h}\in\mathcal{H}} \|\mathbf{h}^* - \mathbf{h}\|_2^2}{\sum_{\mathbf{h}\in\mathcal{H}} \dim(\mathbf{h})}, \quad \mathcal{L}_{\text{Group Ave}} = \frac{1}{|\mathcal{H}|} \sum_{\mathbf{h}\in\mathcal{H}} \frac{\|\mathbf{h}^* - \mathbf{h}\|_2^2}{\dim(\mathbf{h})}. \quad (8)$$

where $\mathbf{h}^*$ and $\mathbf{h}$ denote hidden states of the target model and the initial model, $|\mathcal{H}|$ denotes the number of hidden vectors ($|\mathcal{H}| = 7$ in ResNets), and $\dim(\mathbf{h})$ denotes the dimension number of $\mathbf{h}$. We conduct these ablation studies in order to illustrate the superiority of our logit anchoring method over knowledge distillation (KD) methods or sophisticated hidden state anchoring methods.

## 4.2 MAIN RESULTS

As validated by the main results in Table 1, on both three computer vision and two natural language processing tasks, our proposed anchoring method can achieve competitive backdoor attack success rate and improve both the global and instance-wise consistencies during backdoor learning, especially the instance-wise consistency, compared to the BadNets method and other methods. $L_2$ penalty, EWC, and Surgery methods can also achieve better consistency compared to BadNets under most circumstances. We also adopt the metric ASR+ACC, which denotes the sum of backdoor attack success rate (ASR) and the Top-1 accuracy (ACC), in order to evaluate the comprehensive

Table 2: Backdoor attack success rates (ASR) and consistencies of logit anchoring compared to knowledge distillation (KD) methods and hidden state anchoring methods on CIFAR-10. The clean model is trained on the full dataset, and only 640 images are available in backdoor training. AWP denotes whether perturbations are AWPs. Clean accuracies of small, mid, and large teachers are 93.87%, 94.55%, and 94.58%, respectively.

| Backdoor Attack Method (Setting) | Backdoor ASR (%) | Global Consistency Top-1 ACC (%) | ASR+ACC | Instance-wise Consistency | | | | | Perturbation | |
|---|---|---|---|---|---|---|---|---|---|---|
| | | | | Logit-dis | P-dis | KL-div | mKL | Pearson | $L_2$ | AWP |
| Clean Model (ResNet-34) | - | 94.72 | - | - | - | - | - | - | - | - |
| BadNets | 97.63 | 93.58 | 0 | 1.387 | 0.011 | 0.071 | 0.110 | 0.697 | 1.510 | Y |
| KD (Small Teacher, ResNet-18, $\lambda=2$) | 96.91 | 94.32 | +0.02 | 0.873 | 0.008 | 0.056 | 0.062 | 0.717 | 1.874 | Y |
| KD (Mid Teacher, ResNet-34, $\lambda=2$) | 97.14 | **94.52** | +0.46 | 0.798 | 0.007 | 0.055 | 0.057 | 0.742 | 1.950 | Y |
| KD (Large Teacher, ResNet-101, $\lambda=2$) | 96.84 | 94.33 | -0.04 | 0.813 | 0.007 | 0.059 | 0.061 | 0.739 | 1.887 | Y |
| Hidden Anchoring (All Ave, $\lambda=0.2$) | **97.79** | 93.75 | +0.33 | 1.365 | 0.011 | 0.070 | 0.108 | 0.702 | 1.477 | Y |
| Hidden Anchoring (Group Ave, $\lambda=0.5$) | 97.48 | 94.17 | +0.44 | 0.605 | 0.006 | 0.037 | 0.043 | 0.773 | 1.355 | Y |
| Logit Anchoring (Ours, $\lambda=2$) | 97.28 | 94.41 | **+0.48** | **0.356** | **0.003** | **0.014** | **0.014** | **0.859** | 1.331 | Y |

Table 3: Backdoor attack success rates (ASR) and consistencies of backdoor methods evaluated on the CIFAR-10 dataset with different data accessibility and training settings. AWP denotes whether perturbations are AWPs. As analyzed in Section 3.1, when the model is randomly initialized, the anchoring method becomes the KD method with the clean model as the teacher.

| Settings | Backdoor Attack Method (Setting) | Backdoor ASR (%) | Global Consistency Top-1 ACC (%) | ASR+ACC | Instance-wise Consistency | | | | | Perturbation | |
|---|---|---|---|---|---|---|---|---|---|---|---|
| | | | | | Logit-dis | P-dis | KL-div | mKL | Pearson | $L_2$ | AWP |
| Full data | Clean Model | - | 94.72 | - | - | - | - | - | - | - | - |
| 640 images available | BadNets | 97.63 | 93.58 | 0 | 1.387 | 0.011 | 0.071 | 0.110 | 0.697 | 1.510 | Y |
| | $L_2$ penalty ($\lambda=0.5$) | 93.48 | 93.68 | -4.05 | 1.158 | 0.010 | 0.063 | 0.091 | 0.729 | 0.879 | Y |
| | EWC ($\lambda=0.1$) | 95.20 | 93.81 | -2.20 | 1.420 | 0.011 | 0.059 | 0.098 | 0.739 | 1.420 | Y |
| | Surgery ($\lambda=0.0002$) | **97.67** | 93.89 | +0.35 | 1.207 | 0.009 | 0.055 | 0.082 | 0.752 | 1.449 | Y |
| | Anchoring (Ours, $\lambda=2$) | 97.28 | **94.41** | **+0.48** | **0.356** | **0.003** | **0.014** | **0.014** | **0.859** | 1.331 | Y |
| Full data, initialized with the clean model | BadNets | 99.35 | 94.79 | 0 | 0.998 | 0.007 | 0.051 | 0.064 | 0.739 | 2.109 | Y |
| | $L_2$ penalty ($\lambda=0.1$) | 98.21 | 94.22 | -1.66 | 1.138 | 0.008 | 0.053 | 0.070 | 0.742 | 1.096 | Y |
| | EWC ($\lambda=0.05$) | 98.35 | 94.25 | -1.49 | 1.444 | 0.009 | 0.051 | 0.082 | 0.745 | 2.067 | Y |
| | Surgery ($\lambda=0.0001$) | **99.43** | 94.39 | -0.27 | 0.970 | 0.006 | 0.363 | 0.489 | 0.794 | 1.852 | Y |
| | Anchoring (Ours, $\lambda=0.05$) | 99.32 | **94.93** | **+0.16** | **0.371** | **0.004** | **0.023** | **0.027** | **0.822** | 2.233 | Y |
| Full data, random initialization | BadNets | 99.57 | **94.74** | 0 | 1.428 | 0.012 | 0.193 | 0.186 | 0.547 | 44.10 | N |
| | $L_2$ penalty ($\lambda=0.05$) | 97.89 | 94.50 | -1.92 | 0.932 | 0.008 | **0.050** | 0.060 | 0.718 | 1.448 | Y |
| | EWC ($\lambda=0.001$) | 99.44 | 94.98 | **+0.11** | **0.789** | **0.006** | 0.055 | **0.055** | **0.765** | 9.075 | N |
| | Surgery ($\lambda=0.00002$) | **99.63** | 94.66 | -0.02 | 0.949 | 0.008 | 0.070 | 0.084 | 0.692 | 9.900 | N |
| | KD ($\lambda=0.2$) | 99.44 | 94.40 | -0.47 | 0.681 | 0.009 | 0.111 | 0.108 | 0.674 | 49.47 | N |

performance of the backdoor accuracy (ASR) and the global consistency (ACC) at the same time. For all the methods, we report the gain of ASR+ACC with respect to the ASR+ACC of BadNets. Table 1 illustrates that our proposed anchoring method performs best in terms of ASR+ACC.

**Ablation study.** The results of the ablation study are shown in Table 2. Our proposed anchoring method significantly outperforms the knowledge distillation method in instance-wise consistency. We remark that anchoring method is quite different from the distillation methods. For hidden state anchoring methods, the strong anchoring penalty on all hidden states may harm the learning process, and thus the optimal hyperparameter $\lambda$ is smaller and the instance-wise consistency drops compared to the logit anchoring method. Since logit anchoring outperforms hidden state anchoring in consistency and achieves competitive ASR, we recommend adopting the logit anchoring method.

## 4.3 FURTHER ANALYSIS

Further analysis are conducted on CIFAR-10. Supplementary results are in Appendix.C.

**Results under multiple data settings.** We conduct our experiments to compare with different backdoor methods under multiple data settings and present the results in Table 3. Our proposed anchoring loss can improve both the global and instance-wise consistency during backdoor learning with a small fraction of the dataset or the full dataset. Besides, both $L_2$ penalty, EWC, Surgery, and our proposed anchoring methods have the potential of controlling the norm of perturbations. It means that the theoretical analysis of backdoor learning with adversarial weight perturbation is applicable to their training processes since the perturbations are still small. When the full training

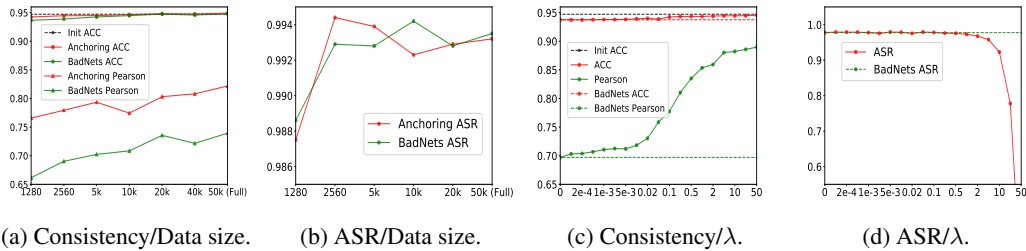

(a) Consistency/Data size.  (b) ASR/Data size.  (c) Consistency/$\lambda$.  (d) ASR/$\lambda$.

Figure 2: Performance of BadNets and anchoring ($\lambda$=0.05) methods with various training data sizes in (a), (b). Performance of anchoring methods with various $\lambda$ (640 images are available) in (c), (d).

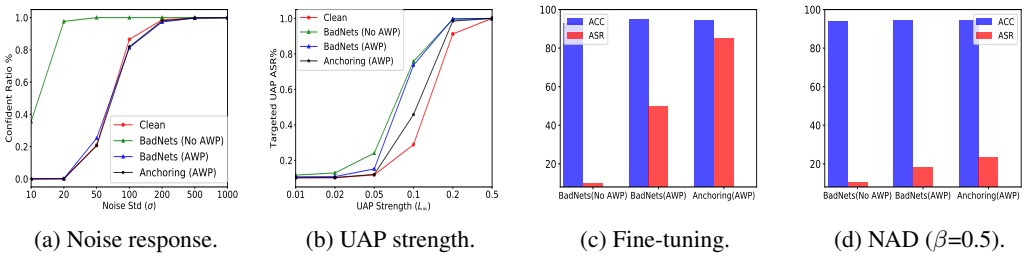

(a) Noise response.  (b) UAP strength.  (c) Fine-tuning.  (d) NAD ($\beta$=0.5).

Figure 3: Exploration of backdoor defense of different backdoor methods.

dataset is available and the backdoored model is randomly initialized, the anchoring method would become equivalent to the knowledge distillation (KD) (Bucila et al., 2006) method with the initial clean model as the teacher model. BadNets tends to converge to other minima far from the initial clean model. $L_2$ penalty can gain a minimal backdoor perturbation due to its inherent learning target. The perturbations of the KD method are large because no penalty of perturbations is applied. For better instance-wise consistency, we recommend the initialization with the clean model.

**Influence of the training data size and the hyperparameter $\lambda$.** We also investigate the influence of the training data size and the hyperparameter $\lambda$. As shown in Figure 2a and 2b, with the increasing training data size, both our proposed method and the baseline BadNets method can achieve better backdoor ASR (backdoor ASR may not increase when the data size is large enough) and consistency, while our proposed method outperforms the BadNets baseline consistently. From Figure 2c and 2d, we can conclude that larger $\lambda$ can preserve more knowledge in the clean model, improve the backdoor consistency but harm the backdoor ASR when $\lambda$ is too large. Therefore, there exists a trade-off between backdoor ASR and consistency, thus we empirically choose a proper $\lambda$, which can achieve the maximum sum of Top-1 ACC and Backdoor ASR.

## 4.4 BACKDOOR DETECTION AND MITIGATION

We implement several backdoor detection and mitigation methods to defend against backdoored models with different backdoor methods on CIFAR-10. Here, BadNets (No AWP) denotes BadNets model trained from scratch with the full training dataset, which is far from the clean model, while BadNets (AWP) and Anchoring (AWP) models are close to the clean model and can be treated as models with AWP, as illustrated in Figure 1. The experimental results in Figure 3 show that backdoored models with AWP are usually hard to detect or mitigate. Our proposed anchoring method can further improve the stealthiness of the backdoor, as its curve is the closeast to the clean model.

**Backdoor detection methods.** Erichson et al. (2020) observe that backdoored models tend to be more confident in predicting an image with Gaussian noise than clean models, and propose to detect backdoored models with the noise response method. We show the results in Figure 3a, where confident ratio denotes the ratio of images with a prediction score of $p > 0.95$. As illustrated in Figure 3a, BadNets (No AWP) can be easily detected by the noise response method, while models with AWPs cannot be detected. We also implement the backdoor detection method via targeted Universal Adversarial Perturbation (UAP) (Moosavi-Dezfooli et al., 2017) with the backdoor target

as the target label following Zhang et al. (2020). From the results in Figure 3b, we can rank the detection difficulty as follows: Anchoring (AWP) > BadNets (AWP) > BadNets (No AWP).

**Backdoor mitigation methods.** Yao et al. (2019) propose to use standard fine-tuning to mitigate backdoor. Li et al. (2021) propose Neural Attention Distillation (NAD) to mitigate backdoor. In our experiment, we split 2000 samples for fine-tuning or NAD, and adopt another clean model as the teacher in NAD. As shown in Figure 3c, the order of the mitigation difficulty is the same as Figure 3b, i.e., Anchoring (AWP) > BadNets (AWP) > BadNets (No AWP). Standard fine-tuning can mitigate backdoor in BadNets (No AWP) completely, mitigate backdoor in BadNets (AWP) partly, but fail to mitigate backdoor in Anchoring (AWP). Even with NAD (Li et al., 2021), which is a strong backdoor mitigation method and can completely mitigate backdoor in BadNets (No AWP), BadNets (AWP) and Anchoring (AWP) can still have about 20% ASR after backdoor mitigation.

## 5 RELATED WORK

**Backdoor Attacks and Defense.** *Backdoor attacks* (Gu et al., 2019) or Trojaning attacks (Liu et al., 2018b) pose serious threats to neural networks, where backdoors may be maliciously injected by data poisoning (Muñoz-González et al., 2017; Chen et al., 2017). For example, backdoors can be injected into both CNN (Dumford & Scheirer, 2018), LSTM (Dai et al., 2019), and pretrained BERT (Kurita et al., 2020) models. *Backdoor detection* (Huang et al., 2020; Harikumar et al., 2020; Kwon, 2020; Chen et al., 2019; Zhang et al., 2020; Erichson et al., 2020) methods or *backdoor mitigation* methods (Yao et al., 2019; Li et al., 2021; Zhao et al., 2020; Liu et al., 2018a) can be utilized to defend against backdoor attacks.

**Reducing Side-effects in Backdoor Learning.** Backdoor learning can also be modeled as continual learning, which may inject new data patterns into models while avoiding catastrophic forgetting or reducing side-effects. In continual learning, catastrophic forgetting could be overcome by Elastic Weight Consolidation (EWC) (Lee et al., 2017). In backdoor learning, side-effects can be reduced via Adversarial Weight Perturbations (AWPs) (Garg et al., 2020) or Targeted Bit Trojans (TBT) (Rakin et al., 2020). Instance-wise side effects can be reduced by neural network surgery (Zhang et al., 2021), which only modifies a small fraction of parameters, and Dumford & Scheirer (2018) also observe that reducing the number of modified parameters can significantly reduce the alteration to the behavior of neural networks. Yang et al. (2021) propose to poison word embeddings of NLP models for minimal side-effects.

## 6 CONCLUSION

In this work, we observe an interesting phenomenon that the variations of parameters are always adversarial weight perturbations (AWPs) when tuning the trained clean model to inject backdoors. We further provide theoretical analysis to explain this phenomenon. We firstly formulate the behavior of maintaining accuracy on clean data as the consistency of backdoored models, including both global consistency and instance-wise consistency. To improve the consistency of the backdoored and clean models on the clean data, we propose a logit anchoring method to anchor or freeze the model behavior on the clean data. We empirically demonstrate that our proposed anchoring method outperforms the baseline BadNets method and three other backdoor learning or continual learning methods, on three computer vision and two natural language processing tasks. Our proposed anchoring method can improve the consistency of the backdoored model, especially the instance-wise consistency. Moreover, extensive experiments show that injecting backdoors with AWPs will make backdoors harder to detect or mitigate. Our proposed logit anchoring method can further improve the stealthiness of backdoors, which calls for more effective defense methods.

## ACKNOWLEDGEMENT

The authors would like to thank anonymous reviewers for their helpful comments. This work is done when Zhiyuan Zhang was a research intern at Ant Group, Hangzhou. Xu Sun and Lingjuan Lyu are corresponding authors.

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

APPENDIX

## A  TECHNICAL DETAILS OF THEORETICAL ANALYSIS

### A.1  TECHNICAL DETAILS OF EXISTENCE OF ADVERSARIAL WEIGHT PERTURBATION

We first prove Proposition 1 and then provide more technical details of Remark 1.

**Proposition 1** (Upper Bound of $\|\boldsymbol{\delta}\|_2$). *Suppose $\boldsymbol{H}$ denotes the Hessian matrix $\nabla_{\boldsymbol{\theta}}^2 \mathcal{L}(\mathcal{D}, \boldsymbol{\theta})$ on the clean dataset, $\mathbf{g}^* = \nabla_{\boldsymbol{\theta}} \mathcal{L}(\mathcal{D}^*, \boldsymbol{\theta})$. Assume $\mathcal{L}(\mathcal{D}^*, \boldsymbol{\theta} + \boldsymbol{\delta}) \leq \mathcal{L}(\mathcal{D}^*, \boldsymbol{\theta}) - |\Delta \mathcal{L}^*|$ can ensure that we successfully inject a backdoor. Suppose we can choose the poisoning ratio, and can control $\eta = |\mathcal{D}^*|/|\mathcal{D}|$, the adversarial weight perturbation $\boldsymbol{\delta}$ ($\boldsymbol{\delta} = \boldsymbol{\delta}(\eta)$ is determined by $\eta$) is,*

$$\boldsymbol{\delta} = \boldsymbol{\delta}(\eta) = -\eta \boldsymbol{H}^{-1}\mathbf{g}^* + o(\|\boldsymbol{\delta}(\eta)\|_2) \tag{9}$$

*and to ensure that we can successfully inject a backdoor, we only need to ensure that $\eta \geq \eta_0$. There exists an adversarial weight perturbation $\boldsymbol{\delta}$,*

$$\|\boldsymbol{\delta}\|_2 \leq \frac{|\Delta \mathcal{L}^*| + o(1)}{\|\mathbf{g}^*\| \cos\langle \mathbf{g}^*, \boldsymbol{H}^{-1}\mathbf{g}^* \rangle} \tag{10}$$

*Proof.* The definition of the loss function on the dataset can be rewritten as,

$$\mathcal{L}(\mathcal{D}; \boldsymbol{\theta}) = \frac{1}{|\mathcal{D}|} \sum_{(\mathbf{x},y)\in\mathcal{D}} [\mathcal{L}((\mathbf{x}, y); \boldsymbol{\theta})], \quad \mathcal{L}(\mathcal{D} \cup \mathcal{D}^*; \boldsymbol{\theta}) = \frac{1}{1+\eta}\big(\mathcal{L}(\mathcal{D}; \boldsymbol{\theta}) + \eta\mathcal{L}(\mathcal{D}^*; \boldsymbol{\theta})\big) \tag{11}$$

According to the definition of optimal parameters,

$$\nabla_{\boldsymbol{\theta}} \mathcal{L}(\mathcal{D}; \boldsymbol{\theta}) = \mathbf{0}, \quad \nabla_{\boldsymbol{\theta}} \mathcal{L}(\mathcal{D}; \boldsymbol{\theta}^*) + \eta \nabla_{\boldsymbol{\theta}} \mathcal{L}(\mathcal{D}^*; \boldsymbol{\theta}^*) = \mathbf{0} \tag{12}$$

$$\big(\nabla_{\boldsymbol{\theta}} \mathcal{L}(\mathcal{D}; \boldsymbol{\theta}^*) - \nabla_{\boldsymbol{\theta}} \mathcal{L}(\mathcal{D}; \boldsymbol{\theta})\big) + \eta \nabla_{\boldsymbol{\theta}} \mathcal{L}(\mathcal{D}^*; \boldsymbol{\theta}^*) = \big(\boldsymbol{H}\boldsymbol{\delta} + o(\|\boldsymbol{\delta}\|_2)\big) + \eta\mathbf{g}^* = \mathbf{0} \tag{13}$$

Therefore, the following equation holds,

$$\boldsymbol{\delta} = -\eta \boldsymbol{H}^{-1}\mathbf{g}^* + o(\|\boldsymbol{\delta}\|_2) \tag{14}$$

Adopting Tayler expansion, the clean loss change and backdoor loss change after injecting backdoor $\boldsymbol{\delta}$ can be written as follows,

$$\mathcal{L}(\mathcal{D}; \boldsymbol{\theta} + \boldsymbol{\delta}) - \mathcal{L}(\mathcal{D}; \boldsymbol{\theta}) = \frac{1}{2}\boldsymbol{\delta}^{\mathrm{T}}\boldsymbol{H}\boldsymbol{\delta} + o(\|\boldsymbol{\delta}\|_2^2) = \frac{\eta^2}{2}\mathbf{g}^{*\mathrm{T}}\boldsymbol{H}^{-1}\mathbf{g}^* + o(\|\boldsymbol{\delta}\|_2^2) \tag{15}$$

$$\mathcal{L}(\mathcal{D}^*; \boldsymbol{\theta} + \boldsymbol{\delta}) - \mathcal{L}(\mathcal{D}^*; \boldsymbol{\theta}) = \boldsymbol{\delta}^{\mathrm{T}}\mathbf{g}^* + o(\|\boldsymbol{\delta}\|_2) = -\eta\mathbf{g}^{*\mathrm{T}}\boldsymbol{H}^{-1}\mathbf{g}^* + o(\|\boldsymbol{\delta}\|_2) \tag{16}$$

To ensure that $\mathcal{L}(\mathcal{D}^*; \boldsymbol{\theta} + \boldsymbol{\delta}) < \mathcal{L}(\mathcal{D}^*; \boldsymbol{\theta}) - |\Delta \mathcal{L}^*|$, we only need to choose $\eta_0 = \frac{|\Delta \mathcal{L}^*|}{\mathbf{g}^{*\mathrm{T}}\boldsymbol{H}^{-1}\mathbf{g}^*} + o(1)$ and ensure $\eta \geq \eta_0$. There exists an adversarial weight perturbation, $\boldsymbol{\delta} = \boldsymbol{\delta}(\eta_0)$ that,

$$\boldsymbol{\delta}(\eta_0) = -\frac{|\Delta \mathcal{L}^*|\boldsymbol{H}^{-1}\mathbf{g}^*}{\mathbf{g}^{*\mathrm{T}}\boldsymbol{H}^{-1}\mathbf{g}^*} + o(\|\boldsymbol{\delta}(\eta_0)\|_2) \tag{17}$$

$$\|\boldsymbol{\delta}(\eta_0)\|_2 = \frac{|\Delta \mathcal{L}^*|\|\boldsymbol{H}^{-1}\mathbf{g}^*\|_2(1 + o(1))}{\|\mathbf{g}^*\|_2\|\boldsymbol{H}^{-1}\mathbf{g}^*\|_2 \cos\langle \mathbf{g}^*, \boldsymbol{H}^{-1}\mathbf{g}^* \rangle} = \frac{|\Delta \mathcal{L}^*| + o(1)}{\|\mathbf{g}^*\| \cos\langle \mathbf{g}^*, \boldsymbol{H}^{-1}\mathbf{g}^* \rangle} \tag{18}$$

Therefore, there exists $\boldsymbol{\delta}$ that can successfully inject a backdoor with $\|\boldsymbol{\delta}\|_2 \leq \|\boldsymbol{\delta}(\eta_0)\|_2$. $\square$

**Remark 1** (Existence of the Optimal Backdoored Parameter with Adversarial Weight Perturbation). *We take a logistic regression model for a two-class classification task as an example. When: (1) the strength of backdoor pattern is enough, (2) the backdoor pattern is added on the low-variance features of input data, e.g., on the blank corner of figures in CV, or choosing a low-frequency word in NLP, then: $\|\boldsymbol{\delta}\|_2$ is small, which ensures the existence of the optimal backdoored parameter with adversarial weight perturbation.*

The formally stated conditions in Remark 1 is that, when: (1) $\|\mathbf{g}^*\|_2$ is large enough, (2) the direction of $\mathbf{g}^*$ is close to that of $\boldsymbol{H}^{-1}\mathbf{g}^*$, namely, the direction of $\mathbf{g}^*$ is close to the eigenvector of $\boldsymbol{H}$ with the minimum eigenvalue, then $\|\boldsymbol{\delta}\|_2$ is small according to Proposition 1.

Here in (2), when the direction of $\mathbf{g}^*$ is close to that of $\boldsymbol{H}^{-1}\mathbf{g}^*$, we may assume $\mathbf{g}^* \approx \mu \boldsymbol{H}^{-1}\mathbf{g}^*$, or $\boldsymbol{H}\mathbf{g}^* \approx \mu \mathbf{g}^*$. Since $\mathcal{L}(\mathcal{D};\boldsymbol{\theta}+\boldsymbol{\delta}) - \mathcal{L}(\mathcal{D};\boldsymbol{\theta}) = \frac{1}{2}\boldsymbol{\delta}^{\mathrm{T}}\boldsymbol{H}\boldsymbol{\delta} \approx \frac{\mu}{2}\|\boldsymbol{\delta}\|_2^2$, we choose $\mu$ as the minimum eigenvalue of $\boldsymbol{H}$ for a smaller clean loss change.

Take a logistic regression model for a two-class classification task as an example, where $y \in \{-1, 1\}$, $p_{\boldsymbol{\theta}}(y|\mathbf{x}) = \sigma(y\boldsymbol{\theta}^{\mathrm{T}}\mathbf{x})$, $\mathcal{L}((\mathbf{x},y);\boldsymbol{\theta}) = \log(1 + \exp(-y\boldsymbol{\theta}^{\mathrm{T}}\mathbf{x}))$ and $\sigma$ is the sigmoid function. The backdoor pattern is to change $(\mathbf{x}, -1)$ into $(\mathbf{x} + \boldsymbol{\Delta}, 1)$. We have,

$$\boldsymbol{H} = \frac{1}{|\mathcal{D}|} \sum_{(\mathbf{x},y)\in\mathcal{D}} (\sigma(\boldsymbol{\theta}^{\mathrm{T}}\mathbf{x})\sigma(-\boldsymbol{\theta}^{\mathrm{T}}\mathbf{x})\mathbf{x}\mathbf{x}^{\mathrm{T}}) \tag{19}$$

$$\mathbf{g}^* = -\frac{1}{|\mathcal{D}^*|} \sum_{(\mathbf{x}+\boldsymbol{\Delta},1)\in\mathcal{D}^*} \sigma(-\boldsymbol{\theta}^{\mathrm{T}}(\mathbf{x}+\boldsymbol{\Delta}))(\mathbf{x}+\boldsymbol{\Delta}) \tag{20}$$

To ensure that (1) $\|\mathbf{g}^*\|_2$ is large enough, we should ensure that the strength of backdoor pattern $\|\boldsymbol{\Delta}\|_2$ is enough. The Hessian matrix is a re-weighted version of $\mathbb{E}[\mathbf{x}\mathbf{x}^{\mathrm{T}}] = \mathbb{E}[\mathbf{x}]\mathbb{E}[\mathbf{x}]^{\mathrm{T}} + \mathbb{D}[\mathbf{x}] = \mathbb{E}[\mathbf{x}]\mathbb{E}[\mathbf{x}]^{\mathrm{T}} + \mathrm{Cov}(\mathbf{x},\mathbf{x})$. To ensure that (2) the direction of $\mathbf{g}^*$ is close to the eigenvector of $\boldsymbol{H}$ with the minimum eigenvalue, we should ensure the direction of $\mathbf{g}^*$ or $\boldsymbol{\Delta}$ is close to the eigenvector of $\boldsymbol{H}$ or $\mathbb{D}(\mathbf{x}) = \mathrm{Cov}(\mathbf{x},\mathbf{x})$ with the minimum eigenvalue. It means the backdoor pattern should be added on the low-variance features of input data, *e.g.*, on the blank corner of figures in CV, or choosing low-frequency words in NLP.

### A.2 DETAILED VERSION OF LEMMA 1 AND FURTHER ANALYSIS.

We introduce Lemma 1 to explain why the optimizer tends to converge to the backdoored parameter $\boldsymbol{\theta}^*$ with adversarial weight perturbation.

**Lemma 1** (Detailed Version. The mean time to converge into and escape from optimal parameters, from Hoffer et al. (2017) and Xie et al. (2021)). *As indicated in Hoffer et al. (2017), the time (step) $t$ for SGD to search an optimal parameter $\boldsymbol{\theta}_1^*$ is $\log t \sim \|\boldsymbol{\theta}_1^* - \boldsymbol{\theta}\|$. As proved in Xie et al. (2021), the mean escape time (step) $\tau$ from the optimal backdoored parameter $\boldsymbol{\theta}^*$ outside the basin near the local minima $\boldsymbol{\theta}^*$ is,*

$$\tau = 2\pi \frac{1}{|H_{be}|} \exp\left(\frac{2B}{\eta_{lr}}\left(\frac{s}{H_{ae}} + \frac{1-s}{|H_{be}|}\right)\Delta\mathcal{L}^*\right) \tag{21}$$

*where $B$ is the batch size, $\eta_{lr}$ is the learning rate, $s \in (0,1)$ is a path-dependent parameter, and $H_{ae}, H_{be}$ denote the eigenvalues of the Hessians at the minima $\boldsymbol{\theta}^*$ and a point outside of the local minima $\boldsymbol{\theta}^*$ corresponding to the escape direction $e$.*

The mean escape time $\tau$ can be rewritten as follows. Suppose $\kappa = H_{ae}$ measures the curvature near the clean local minima $\boldsymbol{\theta}$, $\tau_0 = 2\pi\frac{1}{|H_{be}|}$, $C_1 = 2s$, $C_2 = \frac{2(1-s)}{|H_{be}|}$,

$$\tau = \tau_0 \exp\left(\frac{B}{\eta_{lr}}\left(\frac{C_1}{\kappa} + C_2\right)\Delta\mathcal{L}^*\right), \quad \log\tau = \log\tau_0 + \frac{B}{\eta_{lr}}(C_1\kappa^{-1} + C_2)\Delta\mathcal{L}^* \tag{22}$$

According to Lemma 1, it is easy for optimizers to find the optimal backdoored parameter with adversarial weight perturbation in the early stage of backdoor learning. Modern optimizers have a large $\frac{B}{\eta_{lr}}$ and tend to find flat minima, thus $\kappa$ is small. $\Delta\mathcal{L}^*$ tends to be large since adversarial weight perturbation can successfully inject a backdoor. Therefore, it is hard to escape from $\boldsymbol{\theta}^*$ according to Lemma 1. To conclude, the optimizer tends to converge to the backdoored parameter $\boldsymbol{\theta}^*$ with adversarial weight perturbation.

## A.3  PROOF OF PROPOSITION 2

**Proposition 2.** *In the demo model, suppose* $L = \mathbb{E}_{(\mathbf{x},y)\in\mathcal{D}}[\sum_{i=1}^{C} \epsilon_i^2]$ *denotes the anchoring loss, then,*

$$\Delta\mathcal{L}(\mathcal{D}) = \mathbb{E}_{(\mathbf{x},y)\in\mathcal{D}}\left[\sum_{i,j}\frac{p_i p_j(\epsilon_i - \epsilon_j)^2}{4}\right] + o(L) \leq \mathbb{E}_{(\mathbf{x},y)\in\mathcal{D}}\left[\sum_{i=1}^{C} p_i(1-p_i)\epsilon_i^2\right] + o(L) \quad (23)$$

*Proof.* For the loss function $\mathcal{L}\big((\mathbf{x},y)\big) = -\log p_y$, we calculate the gradients and the Hessian,

$$\frac{\partial p_i}{\partial s_j} = \mathbb{I}(i=j)p_i - p_i p_j, \quad \frac{\partial p_i}{\partial \boldsymbol{\delta}_j} = (\mathbb{I}(i=j)p_i - p_i p_j)\mathbf{x} \quad (24)$$

$$\frac{\partial\mathcal{L}\big((\mathbf{x},y)\big)}{\partial\boldsymbol{\delta}_j} = \frac{\partial(-\log p_y)}{\partial\boldsymbol{\delta}_j} = (-\mathbb{I}(j=y) + p_j)\mathbf{x} \quad (25)$$

$$\frac{\partial^2\mathcal{L}\big((\mathbf{x},y)\big)}{\partial\boldsymbol{\delta}_i\partial\boldsymbol{\delta}_j} = \frac{\partial p_i}{\partial\boldsymbol{\delta}_j}\mathbf{x}^{\mathrm{T}} = (\mathbb{I}(i=j)p_i - p_i p_j)\mathbf{x}\mathbf{x}^{\mathrm{T}} \quad (26)$$

Adopting the second-order Taylor expansion, with the error term $o(\sum_{i=1}^{C}\epsilon_i^2)$,

$$\Delta\mathcal{L}\big((\mathbf{x},y)\big) = \sum_{i,j}\frac{1}{2}\boldsymbol{\delta}_i^{\mathrm{T}}\frac{\partial^2\mathcal{L}}{\partial\boldsymbol{\delta}_i\partial\boldsymbol{\delta}_j}\boldsymbol{\delta}_j + o(\sum_{i=1}^{C}\epsilon_i^2) \quad (27)$$

$$= \frac{\sum_{i=1}^{C} p_i(\boldsymbol{\delta}_i^{\mathrm{T}}\mathbf{x})^2 - \big(\sum_{i=1}^{C} p_i\boldsymbol{\delta}_i^{\mathrm{T}}\mathbf{x}\big)^2}{2} + o(\sum_{i=1}^{C}\epsilon_i^2) = \frac{\sum_{i=1}^{C} p_i\epsilon_i^2 - \big(\sum_{i=1}^{C} p_i\epsilon_i\big)^2}{2} + o(\sum_{i=1}^{C}\epsilon_i^2) \quad (28)$$

$$= \frac{\sum_{i,j} p_j p_i\epsilon_i^2 - \sum_{i,j} p_i\epsilon_i p_j\epsilon_j}{2} + o(\sum_{i=1}^{C}\epsilon_i^2) = \frac{\sum_{i,j} p_j p_i(\epsilon_i^2 + \epsilon_j^2) - 2\sum_{i,j} p_i\epsilon_i p_j\epsilon_j}{4} + o(\sum_{i=1}^{C}\epsilon_i^2) \quad (29)$$

$$= \sum_{i,j}\frac{p_i p_j(\epsilon_i - \epsilon_j)^2}{4} + o(\sum_{i=1}^{C}\epsilon_i^2) \quad (30)$$

$$\quad (31)$$

Calculate the expectation on the dataset $\mathcal{D}$,

$$\Delta\mathcal{L}(\mathcal{D}) = \mathbb{E}_{(\mathbf{x},y)\in\mathcal{D}}\left[\sum_{i,j}\frac{p_i p_j(\epsilon_i - \epsilon_j)^2}{4}\right] + o\big(\mathbb{E}_{(\mathbf{x},y)\in\mathcal{D}}[\sum_{i=1}^{C}\epsilon_i^2]\big) \quad (32)$$

$$\leq \mathbb{E}_{(\mathbf{x},y)\in\mathcal{D}}\left[\sum_{i,j}\frac{p_i p_j(\epsilon_i^2 + \epsilon_j^2)}{2}\right] + o\big(\mathbb{E}_{(\mathbf{x},y)\in\mathcal{D}}[\sum_{i=1}^{C}\epsilon_i^2]\big) = \mathbb{E}_{(\mathbf{x},y)\in\mathcal{D}}\left[\sum_{i,j} p_i p_j\epsilon_i^2\right] + o(L) \quad (33)$$

$$= \mathbb{E}_{(\mathbf{x},y)\in\mathcal{D}}\left[\sum_{i=1}^{C} p_i(1-p_i)\epsilon_i^2\right] + o(L) \quad (34)$$

$$\square$$

## A.4  PROOF OF PROPOSITION 3

**Proposition 3.** *For instance* $(\mathbf{x}, y)$, *the Kullback-Leibler divergence* $KL(p\|p^*)$ *is bounded by the anchoring loss, namely, there exists* $\alpha = \frac{1}{2}$, *that the following inequality holds,*

$$KL(p\|p^*) = \sum_{i=1}^{C} p_i\log\frac{p_i}{p_i^*} \leq (\alpha + o(1))\sum_{i=1}^{C}\epsilon_i^2 \quad (35)$$

*Proof.* Consider the following approximation holds,

$$\frac{\partial p_i}{\partial s_j} = \mathbb{I}(i = j)p_i - p_i p_j \tag{36}$$

$$p_i' - p_i = \sum_j \frac{\partial p_i}{\partial s_j}\epsilon_j = p_i\epsilon_i - \sum_j p_i p_j \epsilon_j + o(\sqrt{\sum_{i=1}^{C}\epsilon_i^2}) = p_i(\epsilon_i - \bar{\epsilon}) + o(\sqrt{\sum_{i=1}^{C}\epsilon_i^2}) \tag{37}$$

where $\bar{\epsilon} = \sum_j p_j \epsilon_j$. Adopting the second-order Taylor expansion, we have the following approximation with the error term $o(\sum_{i=1}^{C}\epsilon_i^2)$,

$$\text{KL}(p||p^*) = \sum_i p_i \log \frac{p_i}{p_i^*} = -\sum_i p_i \log(1 + \frac{p_i^* - p_i}{p_i}) \tag{38}$$

$$= -\sum_i p_i \frac{p_i^* - p_i}{p_i} + \sum_i \frac{(p_i^* - p_i)^2}{2p_i} + o(\sum_{i=1}^{C}\epsilon_i^2) \tag{39}$$

$$= \sum_i \frac{(p_i^* - p_i)^2}{2p_i} + o(\sum_{i=1}^{C}\epsilon_i^2) = \frac{1}{2}\sum_i p_i(\epsilon_i - \bar{\epsilon})^2 + o(\sum_{i=1}^{C}\epsilon_i^2) \tag{40}$$

If $(\alpha + o(1)) \sum_{i=1}^{C}\epsilon_i^2$ is the upper bound of $\text{KL}(p||p^*)$, then $f = \frac{1}{2}\sum_i p_i(\epsilon_i - \bar{\epsilon})^2 - \alpha \sum_{i=1}^{C}\epsilon_i^2 \leq 0$ holds near the zero point $\epsilon_i = 0$,

$$\frac{\partial f}{\partial \epsilon_i} = p_i(\epsilon_i - \bar{\epsilon}) - 2\alpha\epsilon_i, \quad \frac{\partial^2 f}{\partial \epsilon_i \partial \epsilon_j} = -\mathbb{I}(i = j)(2\alpha - p_i) - p_i p_j \tag{41}$$

On the zero point $\epsilon_i = 0$, $\frac{\partial L}{\partial \epsilon_i} = 0$. Therefore, the Hessian matrix $\boldsymbol{H}_f = [\frac{\partial^2 f}{\partial \epsilon_i \partial \epsilon_j}]_{i,j}$ should be semi-negative definite. Suppose $\mathbf{p} = (p_1, p_2, \cdots, p_C)^{\text{T}}$, $\boldsymbol{P} = \text{diag}\{p_1, p_2, \cdots, p_C\}$, $f \leq 0$ is equivalent to the condition that $-\boldsymbol{H}_f = 2\alpha\boldsymbol{I} - \boldsymbol{P} + \mathbf{p}\mathbf{p}^{\text{T}} \succeq \mathbf{0}$.

Since $\mathbf{p}\mathbf{p}^{\text{T}} \succeq \mathbf{0}$ and $\boldsymbol{I} - \boldsymbol{P} \succeq \mathbf{0}$, when $\alpha = \frac{1}{2}$,

$$2\alpha\boldsymbol{I} - \boldsymbol{P} + \mathbf{p}\mathbf{p}^{\text{T}} = (\boldsymbol{I} - \boldsymbol{P}) + \mathbf{p}\mathbf{p}^{\text{T}} \succeq \mathbf{0} \tag{42}$$

$\square$

# B  EXPERIMENTAL SETUPS

We conduct targeted backdoor learning experiments in both the computer vision and natural language processing fields. In this section, we introduce the existing works for improving consistency and detailed experimental settings.

## B.1  INTRODUCTION OF BASELINES AND IMPLEMENTATION SETTINGS.

We introduce the baselines and implementation settings as follows. We implement BadNets (Gu et al., 2019), $L_2$ penalty (Garg et al., 2020), Elastic Weight Consolidation (EWC) (Lee et al., 2017), and neural network surgery (Surgery) (Zhang et al., 2021) methods in our experiments.

### B.1.1  BADNETS.

The learning target of BadNets (Gu et al., 2019) is:

$$\boldsymbol{\theta}^* = \arg\min \mathcal{L}(\mathcal{D} \cup \mathcal{D}^*; \boldsymbol{\theta}^*) \tag{43}$$

### B.1.2 PENALTY.

Garg et al. (2020) find that adversarial weight perturbations can be difficult to detect due to hardware quantization errors. Therefore, they propose to minimize $\|\boldsymbol{\theta}^* - \boldsymbol{\theta}\|_p$ at the meantime when injecting backdoors. They formulate the optimization target as,

$$\boldsymbol{\theta}^* = \arg \min_{\|\boldsymbol{\theta}^* - \boldsymbol{\theta}\|_p \le \epsilon} \mathcal{L}(\mathcal{D} \cup \mathcal{D}^*; \boldsymbol{\theta}^*) \tag{44}$$

and they solve the optimization via the PGD algorithm. We also conduct experiments of their implementation in Appendix.C.3.

Our implementation to minimize $\|\boldsymbol{\theta}^* - \boldsymbol{\theta}\|_p$ during backdoor learning is adding an $L_2$ penalty on the loss function (Lee et al., 2017; Garg et al., 2020). In our work, we formulate the learning target as the $L_2$ Lagrange relaxation loss or $L_2$ penalty (Garg et al., 2020; Lee et al., 2017):

$$\boldsymbol{\theta}^* = \arg \min \frac{1}{\lambda + 1} \mathcal{L}(\mathcal{D} \cup \mathcal{D}^*; \boldsymbol{\theta}^*) + \frac{\lambda}{\lambda + 1} \|\boldsymbol{\theta}^* - \boldsymbol{\theta}\|_2^2 \tag{45}$$

### B.1.3 EWC.

Lee et al. (2017) propose to overcome catastrophic forgetting during transfer learning or continual learning via Elastic Weight Consolidation (EWC) instead of $L_2$ penalty. The optimization target of EWC in backdoor learning can be written as,

$$\boldsymbol{\theta}^* = \arg \min \frac{1}{\lambda + 1} \mathcal{L}(\mathcal{D} \cup \mathcal{D}^*; \boldsymbol{\theta}^*) + \frac{\lambda}{\lambda + 1} \sum_{i=1}^{n} F_i (\theta_i^* - \theta_i)^2 \tag{46}$$

where $F_i$ denotes the $i$-th element in the diagonal of Fisher information matrix, which is an approximation of $H_{ii}$ in Hessian matrix $\boldsymbol{H}$ on $\mathcal{D}$. The EWC regularization term $\frac{1}{2} \sum_{i=1}^{n} F_i (\theta_i^* - \theta_i)^2$ can be treated as an approximation of clean loss change, $\frac{1}{2} \sum_{i=1}^{n} F_i (\theta_i^* - \theta_i)^2 \approx \frac{1}{2} \sum_{i=1}^{n} H_{ii} (\theta_i^* - \theta_i)^2 \approx \frac{1}{2} \boldsymbol{\delta}^{\mathrm{T}} \boldsymbol{H} \boldsymbol{\delta} \approx \mathcal{L}(\mathcal{D}; \boldsymbol{\theta} + \boldsymbol{\delta}) - \mathcal{L}(\mathcal{D}; \boldsymbol{\theta})$.

$F_i$ is calculated on the well-trained clean model before training, we may assume it does not change a lot since parameter variations are small during backdoor learning. Besides, we normalize and smooth $F_i$ for better stability:

$$F_i' = \gamma \frac{F_i}{\sum_{i=1}^{n} F_i} + (1 - \gamma) \tag{47}$$

where we choose $\gamma = 0.9$.

### B.1.4 SURGERY.

Zhang et al. (2021) propose to inject backdoors with only a small fraction of parameters changed compared to a well-trained clean model, and adopt the $L_1$ Lagrange relaxation loss to enforce models to satisfy $\{\boldsymbol{\delta} : \|\boldsymbol{\delta}\|_0 \le k\}$:

$$\boldsymbol{\theta}^* = \arg \min \frac{1}{\lambda + 1} \mathcal{L}(\mathcal{D} \cup \mathcal{D}^*; \boldsymbol{\theta}^*) + \frac{\lambda}{\lambda + 1} \|\boldsymbol{\theta}^* - \boldsymbol{\theta}\|_1 \tag{48}$$

For more sparse $\boldsymbol{\delta}$, Zhang et al. (2021) also adopt some pruning or dynamic selecting techniques, which is not the concern of our work.

### B.2 DETAILED EXPERIMENTAL SETTINGS

In all experiments, we report the results of the model on the epoch with maximum ACC+ASR. In our implementation, $\mathcal{D}$ can be a small training dataset or the full training dataset, $\mathcal{D}^*$ is poisoned from $\mathcal{D}$, and the poisoning ratio is 50%.

### B.2.1 COMPUTER VISION

The initial model is ResNet-34 (He et al., 2016). We conduct image recognition tasks on CIFAR-10 (Torralba et al., 2008), CIFAR-100 (Torralba et al., 2008), and Tiny-ImageNet (Russakovsky et al., 2015). When training the initial model, the learning rate is 0.1, the weight decay is $5 \times 10^{-4}$ and momentum is 0.9, and batch size is 128. The optimizer is SGD. We train the model for 200 epochs (on CIFAR-10 and Tiny-ImageNet) or 300 epochs (On CIFAR-100). After 150 epochs and 250 epochs, the learning rate is divided by 10.

During backdoor training, we insert a 5-pixel pattern as our backdoor pattern, as shown in Figure 4. Images with backdoor patterns are targeted to be classified as class 0. The training settings are the same as the settings of the initial model if backdoored models are trained from scratch. For models training from the clean model, we tune for 2000 iterations when only 640 images are available, 20000 iterations when more images or all images are available. The learning rate is 0.001. Other settings are the same as the initial model. We grid search hyperparameter $\lambda$ in {1e-5, 2e-5, 5e-5, 1e-4, 2e-4, 5e-4, 1e-3, 2e-3, 5e-3, 0.01, 0.02, 0.05, 0.1, 0.2, 0.5, 1, 2, 5, 10, 20, 50, 100}.

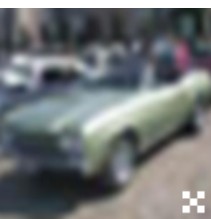

Figure 4: An illustration of the 5-pixel backdoor pattern on CIFAR-10.

### B.2.2 NATURAL LANGUAGE PROCESSING

The initial model is a fine-tuned BERT (Devlin et al., 2019). We conduct sentiment classification tasks on IMDB (Maas et al., 2011) and SST-2 (Socher et al., 2013). The text is uncased and the maximum length of tokens is 384. The fine-tuned BERT is the uncased BERT base model (Devlin et al., 2019). The optimizer is the AdamW optimizer with the training batch size of 8 and the learning rate of $2 \times 10^{-5}$. We fine-tune the model for 10 epochs.

During backdoor training, our trigger word is a low-frequency word "cf". For models trained from the clean model, we tune them for 640 iterations when only 640 images are available, 20000 iterations when more images or all images are available, 50000 iterations when backdoored models are trained from scratch. Other settings are the same as the initial model. During hyperparameter selection, we grid search $\lambda$ in {1e-5, 2e-5, 5e-5, 1e-4, 2e-4, 5e-4, 1e-3, 2e-3, 5e-3, 0.01, 0.02, 0.05, 0.1, 0.2, 0.5, 1, 2, 5, 10}.

### B.3 CHOICE OF METRICS TO EVALUATE INSTANCE-WISE CONSISTENCY

Zhang et al. (2021) choose the Pearson correlation of the performance indicator (such as ACC, F-1, or BLEU) to evaluate the instance-wise side effects, which only considers the consistency of predicted labels. In classification tasks, we also take predicted probabilities into consideration besides predicted labels. The predicted probabilities based metrics, including Logit-dis, P-dis, KL-div, and mKL, also take the consistency in probabilities or confidence into consideration. Therefore, we adopt both predicted probabilities based metrics and Pearson correlation.

### B.4 DEFENSE DETAILS

In fine-tuning (Yao et al., 2019) or NAD (Li et al., 2021) defense, only 2000 instances are available. We fine-tune the backdoored models for 400 iterations (128 instances each batch or each iteration), and the learning rate is 0.005. Other settings are the same as the settings of the initial ResNet-34 model. The teacher model in NAD is anthoer ResNet-34 model.

## C  Supplementary Experimental Results

### C.1  Influence of the hyperparameter

We also investigate the influence of the hyperparameter $\lambda$ on $L_2$ penalty methods. Detailed results are shown in Figure 5. On both the $L_2$ penalty and our proposed anchoring methods larger $\lambda$ can preserve more knowledge in the clean model, improve the backdoor consistency but harm the backdoor ASR when $\lambda$ is too large. Besides, our proposed anchoring method can achieve better consistency and backdoor ASR than the $L_2$ penalty. Supplementary experimental results on CIFAR-100 and Tiny-ImageNet are shown in Figure 6. Similar conclusions can be drawn from Figure 6.

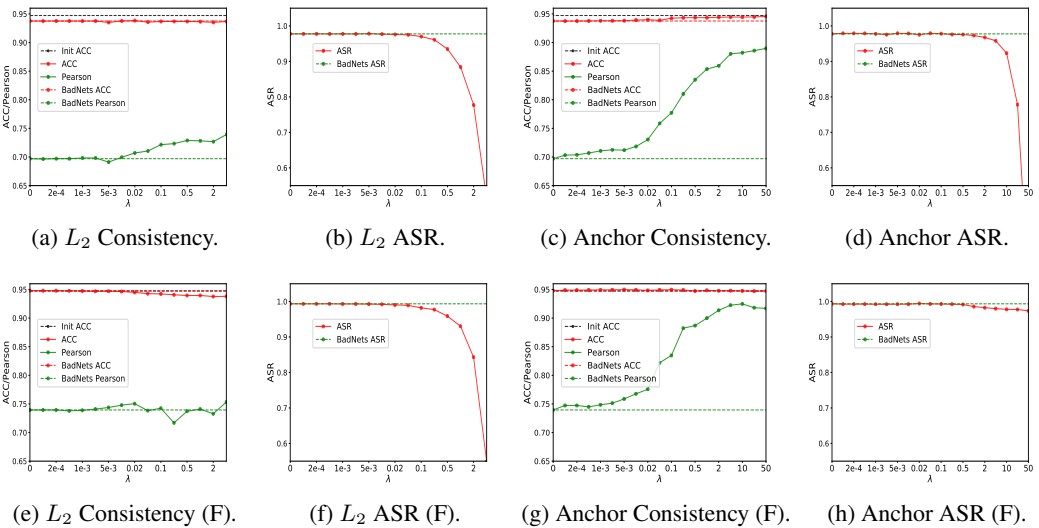

Figure 5: Performance of $L_2$ penalty and anchoring methods with various $\lambda$ on CIFAR-10. Here $L_2$ means the $L_2$ penalty, Anchor means the anchoring method, and F means the full dataset is available, otherwise, only 640 images are available.

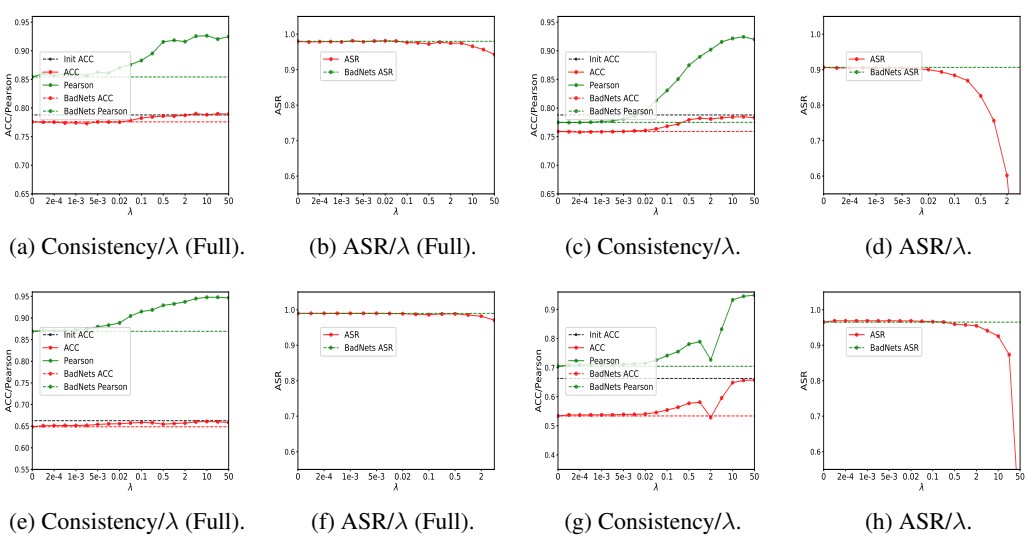

Figure 6: Performance of the anchoring method with various $\lambda$ on CIFAR-100 (a-d) and Tiny-ImageNet (e-h). Here Full means the full dataset is available, otherwise 640 images are available.

### C.2 Visualization of the instance-wise consistency of different anchoring backdoor methods

We also visualize the instance-wise logit consistency of our proposed anchoring method and the baseline BadNets method in Figure 7 (on CIFAR-10), Figure 8 (on CIFAR-100), and Figure 9 (on Tiny-ImageNet). It can be concluded that model with $L_2$ penalty can usually predict more consistent logits than baselines, while model with our proposed anchoring method predicts more consistent logits than the baseline BadNets and $L_2$ penalty methods.

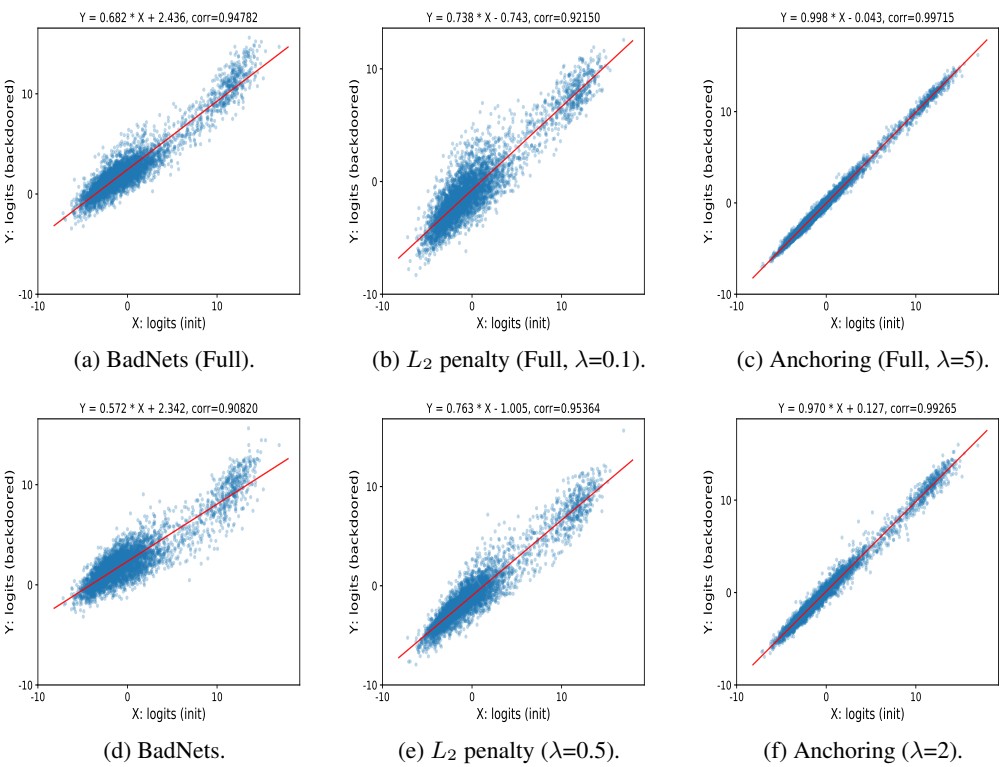

Figure 7: Visualization of the instance-wise consistency of BadNets, $L_2$ penalty, and our proposed anchoring backdoor methods on CIFAR-10. Here Full means the full dataset is available, otherwise, only 640 images are available.

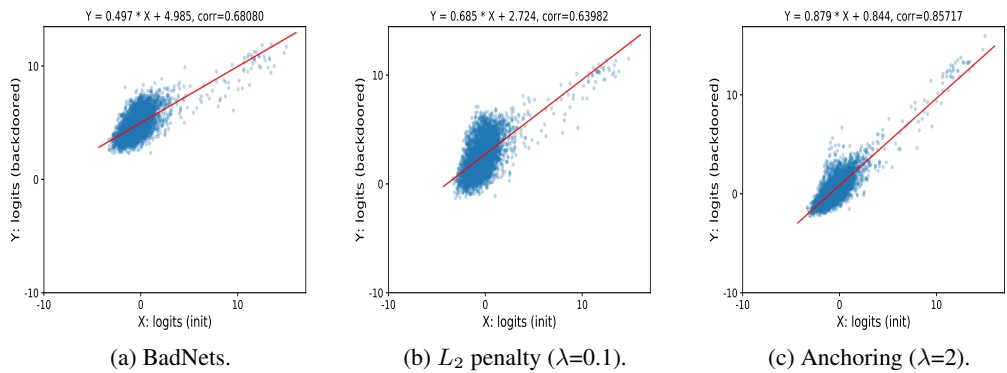

Figure 8: Visualization of the instance-wise consistency of BadNets, $L_2$ penalty, and our proposed anchoring backdoor methods on CIFAR-100. Here only 640 images are available.

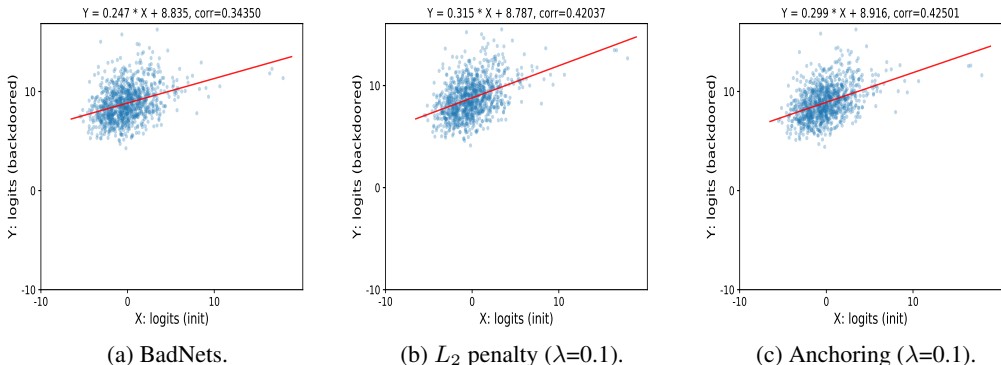

(a) BadNets.  (b) $L_2$ penalty ($\lambda$=0.1).  (c) Anchoring ($\lambda$=0.1).

Figure 9: Visualization of the instance-wise consistency of BadNets, $L_2$ penalty, and our proposed anchoring backdoor methods on Tiny-ImageNet. Here only 640 images are available.

### C.3 DISCUSSION OF OTHER METHODS FOR IMPROVING CONSISTENCY

We also conduct experiments of other methods that may improve consistency on CIFAR-10 and results are reported in Table 4. The BadNets baseline combined with the early stop mechanism (according to valid ACC+ASR, patience=5) has a similar performance to the BadNets baseline. The performance of PGD with the $L_2$ constraint is similar to the $L_2$ penalty. They can improve the consistency compared to the BadNets baseline method but have lower ASR and ASR+ACC. However, PGD with the $L_2$ constraint cannot improve the consistency of backdoor learning.

Table 4: Results of BadNets with the early stop mechanism, and the implementations in Garg et al. (2020) with the PGD algorithm with $L_2$ or $L_{+\infty}$ constraint on CIFAR-10.

| Backdoor Attack Method (Setting) | Backdoor ASR (%) | Global Consistency Top-1 ACC (%) | ASR+ACC | Instance-wise Consistency | | | | |
|---|---|---|---|---|---|---|---|---|
| | | | | Logit-dis | P-dis | KL-div | mKL | Pearson |
| Clean Model (Full data) | - | 94.72 | - | - | - | - | - | - |
| BadNets | 97.63 | 93.58 | 0 | 1.387 | 0.011 | 0.071 | 0.110 | 0.697 |
| BadNets+Early Stop | 97.45 | 93.68 | -0.08 | 1.441 | 0.011 | 0.072 | 0.116 | 0.707 |
| $L_2$ penalty ($\lambda$=0.5) | 93.48 | 93.68 | -4.05 | 1.158 | 0.010 | 0.063 | 0.091 | 0.729 |
| PGD ($L_2, \epsilon = 0.8$) | 90.93 | 93.76 | -6.52 | 1.161 | 0.009 | 0.060 | 0.082 | 0.728 |
| PGD ($L_2, \epsilon = 0.9$) | 94.83 | 93.75 | -2.63 | 1.104 | 0.010 | 0.058 | 0.083 | 0.737 |
| PGD ($L_{+\infty}, \epsilon = 0.0008$) | 94.79 | 92.79 | -3.63 | 1.583 | 0.015 | 0.108 | 0.182 | 0.662 |
| PGD ($L_{+\infty}, \epsilon = 0.001$) | 96.35 | 92.95 | -1.91 | 1.568 | 0.014 | 0.098 | 0.162 | 0.667 |
| EWC ($\lambda$=0.1) | 95.20 | 93.81 | -2.20 | 1.420 | 0.011 | 0.059 | 0.098 | 0.739 |
| Surgery ($\lambda$=0.0002) | **97.67** | 93.89 | +0.35 | 1.207 | 0.009 | 0.055 | 0.082 | 0.752 |
| Anchoring (Ours, $\lambda$=2) | 97.28 | **94.41** | **+0.48** | **0.356** | **0.003** | **0.014** | **0.014** | **0.859** |

### C.4 DETAILED VISUALIZATION OF LOSS BASIN

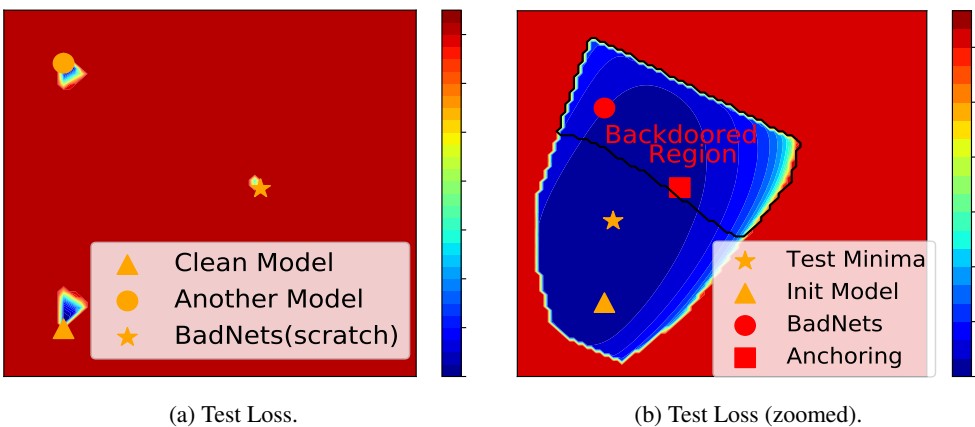

(a) Test Loss.  (b) Test Loss (zoomed).

Figure 10: Detailed visualization of the loss basin.

