# OpenReview forum: "How to Inject Backdoors with Better Consistency: Logit Anchoring on Clean Data"
_ICLR.cc/2022/Conference — ICLR 2022 Poster_

### Official Review · Reviewer_oesM · 2021-10-17

**Correctness:** 4
**Technical Novelty And Significance:** 2
**Empirical Novelty And Significance:** 3
**Recommendation:** 6
**Confidence:** 3

**Main Review:**

The authors point out that “the surge in the usage of the large-scale neural networks makes it hard to train backdoored models from scratch, since it requires a large training dataset.”  However, works which propose to backdoor models trained from scratch are typically putting poisoned data into the victim’s dataset and are not training the model themselves.  This is not to say that the authors are incorrect or anything, but they might want to frame this sentence in the context of this existing work so it doesn’t sound out of place.  Also, all the examples in this paper are ones in which training from scratch, even for a poisoner, is feasible computationally, so this motivation is also a bit off in that sense too.

The method is extremely simple, and I think the theory is a bit extraneous.  The paper could present the method much simpler and more efficiently.

Table 1 results are inconclusive to me.  I’m not sure how important the “instance-wise consistency” gains are, and I think the poisoner usually doesn’t mind if the victim’s accuracy drops by a small degree.  Nonetheless, I am inclined to think that the gains in victim accuracy could have some value and are quite consistent, despite incurring only a very minor cost in ASR.

**Summary Of The Paper:**

This work proposes a logit anchoring loss term for fine-tuning models to introduce backdoors while maintaining consistency on clean data.  Their empirical results show that this term does in fact improve the consistency on clean data and helps maintain the clean accuracy of backdoored models.

**Summary Of The Review:**

The motivation of this work seems a little tenuous, and the presentation is overly complicated.  Nonetheless, the experiments do a good job of showing that their method accomplishes their goal.

---

> ### Author Response · Authors · 2021-11-20
> **Responses to Reviewer oesM**
>
> Thank you for your insightful review. Here are responses to your comments in Main Review:
>
> **Q1** (In Para. 1). The authors point out that “the surge in the usage of the large-scale NN makes it hard to train backdoored models from scratch since it requires a large training dataset”. The motivation of fine-tuning models to introduce backdoors is tenuous. All examples in this paper are ones in which training from scratch is feasible computationally.
>
> **A1**. The poisoner tends to fine-tune backdoored models instead of training from scratch when the original clean training set is not available. The poisoner might create a small dataset for fine-tuning. However, creating a large training set and training from the scratch is costly and difficult. Also, the inconsistency between the original dataset and the created dataset will lead to drops in the clean ACC and instance-wise consistency, too.
>
> To make it clearer, we revise the sentence to: “On the other hand, it is hard to train backdoored models from scratch when the training dataset is sensitive and not available.”
>
> Also, in the examples of this paper, we assume the poisoner only has a small fraction of the clean data (maybe created or annotated by the poisoner), thus the poisoner tunes the trained models instead of training from scratch.
>
>
> **Q2** (In Para. 2). The method is simple.
>
> **A2**. Our main idea is to anchor the model behaviors on the clean data for better consistency in backdoor learning and propose the logit anchoring method. In the ablation study, we find that other sophisticated variants such as anchoring hidden states may harm the learning process, thus we recommend the simple logit anchoring method.
>
>
> **Q3** (In Para. 2). The presentation is complicated. The paper could present the method much simpler.
>
> **A3**. The contribution of the paper includes both theoretical analysis and the proposed method for better consistency. Thus, we choose to present our theoretical analysis first to support our motivation and then present our method. To make the presentation clearer, we only introduce the informal version of complicated theorems in the main paper and give more details in Appendix A in the revised version.
>
>
> **Q4** (In Para. 3). How important the “instance-wise consistency” gains are?
>
> **A4**. Existing studies of backdoor learning usually focus on the clean accuracy and backdoor ASR. However, we propose that an ideal backdoor should also have good instance-wise consistency on clean data. Backdoors with better instance-wise consistency can help ensure that the behaviors of models on the clean data are not altered severely, and they are hard to detect or mitigate.

---

### Official Review · Reviewer_RJ1F · 2021-10-23

**Correctness:** 3
**Technical Novelty And Significance:** 4
**Empirical Novelty And Significance:** 4
**Recommendation:** 8
**Confidence:** 4

**Main Review:**

Strengths:
(1) The theoretical analysis provides interesting insights in injecting backdoors with AWP. Proposition 1 estimates the upper bound norm of the AWP and gives a theoretical guarantee of AWP. Remark 1 is also very interesting, where the authors try to explain why ordinary backdoor methods choose to add triggers on the low-frequency features, because it tends to gain a higher $cos<g^*, H^{-1}g^*>$, thus gain a lower $\|\delta\|_2$ and make it easy to inject backdoors.
(2) The authors propose a novel and interesting concept of consistency in backdoor learning, and propose to evaluate them with several metrics.
(3) The proposed logit anchoring method is simple yet effective, and the experiments are solid. The authors conduct experiments on three CV tasks with the ResNet model and two NLP tasks with the BERT model. They also compare with the methods anchoring other layers instead of the output logits, and compare with other knowledge distillation methods. The ablation study and further analysis are comprehensive, and provide new insights for the future study in this domain.

Weakness:
(1) The authors state that they are the first to propose the concept of consistency, including global and instance-wise consistency. The paper would benefit from examples of instance-wise consistency, which can illustrate why instance-wise consistency is also important besides global consistency. It would be helpful for the paper to be contextualized in existing discussions of instance-wise consistency and the introduction of existing methods.
(2) The introduction and implementation details of existing methods are not well clarified. How do you calculate the Hessian F_{i} or H_{ii} in EWC [2]? I suppose that calculating them in each iteration is very costly. Besides, the authors introduce existing efforts in Appendix B.1, and repeat the formulations in B.2. What is the key difference between B.1 and B.2? I suggest that the authors merge Appendix B.1 and B.2.

Question:
(1) $L_2$-penalty [1] and EWC [2] can also be treated as a regularizer in backdoor learning. I doubt whether other regularizers can also improve the consistency of backdoor learning, even if these regularizers are not initially designed for better consistency. If so, what is the difference for methods improving consistency and regularizers. For examples, can an early stop mechanism or a gradient clip regularizer improve the consistency?
(2) [1] propose to inject backdoors into clean models via AWP. The authors implement it via adding an $L_2$-penalty regularizer, namely minimizing $\min loss+\lambda\|\delta\|_2^2$. It can be also implemented with solving a constrained optimization problem, namely minimizing $\min loss, s.t. \|\delta \|_p\le \epsilon (p=2 or p=+infinity)$ with PGD. Are experimental results of [1] similar with an $L_2$-penalty regularizer or with solving a constrained optimization problem? Why the authors choose an $L_2$-penalty as the implementation?

[1] Siddhant Garg, Adarsh Kumar, Vibhor Goel, and Yingyu Liang. Can adversarial weight perturbations inject neural backdoors.
[2] Sang-Woo Lee, Jin-Hwa Kim, Jaehyun Jun, Jung-Woo Ha, and Byoung-Tak Zhang. Overcoming catastrophic forgetting by incremental moment matching.

**Summary Of The Paper:**

In this work, the authors first analyze the behavior of injecting backdoors into a well-trained clean model via fine-tuning it on a poisoned dataset. The authors point out that only evaluating the backdoor performance with ASR and ACC is not enough since ACC can only evaluate the global consistency, and propose to evaluate the consistency of the backdoor performance with both global and instance-wise consistency. In order to achieve better consistency, they propose a novel anchoring loss to anchor or freeze the model behaviors on the clean data, with a theoretical guarantee. Both the analytical and empirical results validate the effectiveness of their anchoring loss in improving the consistency, especially the instance-wise consistency.

**Summary Of The Review:**

In this work, the authors propose a novel and interesting concept of consistency in backdoor learning, and propose to evaluate them with several metrics. They propose a simple yet effective logit anchoring method for better consistency. Extensive experiments are conducted on three CV tasks with the ResNet model and two NLP tasks with the BERT model. The experiments are very solid. Overall, I think this work opens a new angle of understanding model behaviours in backdoor learning, and I would recommend to strongly accept this paper.

---

> ### Author Response · Authors · 2021-11-20
> **Responses to Reviewer RJ1F**
>
> Thank you for your positive review. Here are responses to your comments:
>
> **Q1**. The introduction and implementation details of existing methods are not well clarified. How do you calculate the Hessian $F_{i}$ or $H_{ii}$? I suppose that calculating them for every iteration is very costly. … I suggest that the authors merge Appendix B.1 and B.2.
>
> **A1**. We calculate Hessian $F_{i}$ on the clean data on a well-trained clean model and do not change it during backdoor learning, since the variation in backdoor learning is AWP. For a clearer presentation, we merge Appendix B.1 and B.2, add more implementation details and introduce existing methods in more detail in Appendix.
>
> **Q2**. What is the difference between methods improving consistency and regularizers? For example, can an early stop mechanism or a gradient clip regularizer improve the consistency?
>
> **A2**. Some regularizers might also improve the consistency though they are not designed for it, such as $L_2$ or EWC. We also try a typical early stop mechanism and show that the BadNets baseline combined with the early stop mechanism has a similar performance to the BadNets baseline, and the results are reported in Appendix C.3 in the revised version.
>
> **Q3**. AWP can be also implemented with solving a constrained optimization problem, namely minimizing $\min \text{loss}, s.t. \|\delta \|_p\le \epsilon\  (p=2\ or\ p=+\infty)$ with PGD.
>
> **A3**. We implement the PGD algorithm with $L_2$ and $L_\infty$ constraints, the results are reported in Appendix C.3 in the revised version. The performance of PGD with the $L_2$ constraint is similar to the $L_2$ penalty implementation. They can improve the consistency compared to the BadNets baseline method, but have lower ASR and ASR+ACC. However, PGD with the $L_\infty$ constraint cannot improve the consistency of backdoor learning.

---

> > ### Comment · Reviewer_RJ1F · 2021-11-26
> > **Keep the score**
> >
> > I have read the response of the authors. Some details and concerns have been addressed. Overall, it is a good work and I vote for acceptance.

---

### Official Review · Reviewer_GfFx · 2021-10-24

**Correctness:** 4
**Technical Novelty And Significance:** 4
**Empirical Novelty And Significance:** 4
**Recommendation:** 8
**Confidence:** 4

**Main Review:**

Reasons to Accept:

(1) The paper is well-written.

(2) The authors propose a logit anchoring method. To validate the effectiveness of the proposed logit anchoring methods, they conduct extensive experiments on three CV tasks and two NLP tasks. The authors also consider many alternative anchoring methods and knowledge distillation methods,  and conduct detailed ablation studies. They also investigate the influence of training set size and the hyper-parameters. Overall, the experimental results and analysis are very solid.

(3) Theoretical insight gained from the paper is novel and inspiring. It explains why a small variation in parameters during fine-tuning can inject backdoors.

Reasons to Reject:

(1) The Surgery method also investigates the instance-wise side effects, the difference between this work and Surgery should be elaborated.

Questions to the authors:

(1) Why backdoor models with AWPs are hard to detect or mitigate? Further discussion is expected.

(2) Why EWC, Surgery or logit anchoring can improve the global consistency of the backdoored model for a big gap on IMDB compared to the BadNets baseline, while on other tasks the gap is much smaller? Is it difficult to inject backdoors into BERT model on the IMDB dataset?

(3) Some theoretical results may be hard to understand. In proposition 1, in Eq 1, \delta=-\eta*H^{-1}g^*+o(||\delta||_2): \delta is a vector, and o(||\delta||_2) is an infinitely small real number, what does it mean by \delta = a vector + a real infinitely small number? Does the plus sign here mean the element-wise plus?

(4) Minors: In Appendix. C, figures and their corresponding texts are too far, which might be puzzling. The presentation of the supplementary material could be improved. For example, the layout of the figures in appendix could be improved. P20 and P21 is a bit empty.




**Summary Of The Paper:**

The paper proposes a novel logit anchoring method to improve the consistency between clean models and backdoored models. The authors state that they are the first to formulate the behavior of maintaining accuracy on clean data as the consistency of backdoored models. They extensively analyze the effects of AWPs on the consistency of backdoored models. Both the theoretical and empirical results validate the effectiveness of anchoring loss in improving consistency, especially the instance-wise consistency.

**Summary Of The Review:**

The paper proposes a novel logit anchoring method to improve the consistency between clean models and backdoored models. The paper formulates the concept of consistency. However, it may benefit from a discussion of the difference between the concept consistency and the concept instance-wise side-effects proposed by the existing work, such as Neural Network Surgery. Overall, this paper is well-written. It evaluates the lower bound of the parameter variation in the backdoor learning and explains the phenomenon of AWP. The experiments are solid. The authors also conduct many extensive analysis and ablation studies. Thus the paper is comprehensive, therefore I am glad to recommend a strong acceptance of this paper.

---

> ### Author Response · Authors · 2021-11-20
> **Responses to Reviewer GfFx**
>
>
> Thank you for your detailed review. Here are responses to your comments:
>
> **Q1**. What is the novelty or contribution of this paper compared to Surgery?
>
> **A1**. The neural network surgery method proposes to modify only a small fraction of parameters to minimize side effects and help reduce the transportation cost for backdoor parameters. Our proposed method improves the consistency via a logit anchoring method, which takes both the instance-wise consistency and the global consistency (overall accuracy) into consideration. Extensive experiments validate the effectiveness of our proposed method. We also provide theoretical insights to explain the AWP phenomenon and the relation between the AWP and consistency. Moreover, we propose a novel framework and some technical indicators to evaluate the consistency.
>
> **Q2**. Why backdoor models with AWPs are hard to detect or mitigate?
>
> **A2**. We conjecture this is because backdoors injected with AWPs are hard to detect since the variations of parameters are small and the behavior is more similar to AWPs. Also, mitigating backdoors with AWPs under the guidance of the clean data is difficult, since the consistency of backdoors with AWPs is higher than backdoors training from scratch.
>
> **Q3**. Why EWC, Surgery, or logit anchoring can improve the global consistency of the backdoored model for a big gap on IMDB compared to the BadNets baseline, while on other tasks the gap is much smaller?
>
> **A3**. This is because injecting backdoors into BERT on the IMDB dataset is difficult. Sentences in IMDB (average ~234) is much longer than sentences in SST-2 (average ~10). Therefore, the ACC of BadNets on IMDB is much lower than that on SST-2, both with 64 sentences.
>
> **Q4**. Some theoretical results may be hard to understand. In proposition 1, in Eq 1, $\delta=-\eta H^{-1}g^\*+o(|\delta|_2)$.
>
> **A4**. It means that $\delta$ can be estimated with $-\eta H^{-1}g^\*$, and the length of the error vector is a higher-order infinitely compared to $||\delta||_2$. Namely,  $\|error\|_2=\|\delta-(-\eta H^{-1}g^\*)\|_2=o(|\delta|_2)$.
>
> **Q5**. Minors.
>
> **A5**. We have improved our presentations in the revised version and the layout is more reasonable now.

---

> > ### Comment · Reviewer_GfFx · 2021-11-21
> > **Keep My Original Score**
> >
> > Thanks to the authors for addressing all my concerns. Given the discussions, I will keep my original score and recommend strong acceptance.

---

### Official Review · Reviewer_rNcw · 2021-10-29

**Correctness:** 4
**Technical Novelty And Significance:** 4
**Empirical Novelty And Significance:** 4
**Recommendation:** 8
**Confidence:** 4

**Main Review:**

Strengths:
(1) The theoretical insights of the paper: The paper considers a specific yet common circumstance of backdoor learning, injecting backdoors into a trained clean model, and observes that variations of parameters are always AWPs. They provide a theoretical explanation, and the analysis provides some interesting theoretical insights of injecting backdoors via fine-tuning a trained clean model.
(2) The methodological contribution: They compare their proposed logit anchoring method with typical existing works and experiments cover both CV and NLP tasks. The experiments are solid and the analysis and ablation studies are comprehensive.

Weakness:
(1) The reasons of choosing five metrics to evaluate the instance-wise consistency need more discussion. Details are in Question (1).

Question:
(1) The paper formulates the consistency of backdoored models, including global consistency (Clean Acc) and instance-wise consistency. The authors propose five metrics to evaluate the instance-wise consistency. Experiments show that the evaluation results of these indicators are consistent in most cases. Is it possible to use other metrics, such as Pearson correlation?

**Summary Of The Paper:**

This paper focuses on injecting backdoors into a trained clean model. The authors provide a theoretical analysis to illustrate why the variations are always AWP or small weight perturbations in this circumstance. The concept of consistency is firstly formulated, and the authors propose a logit anchoring method to improve the consistency, especially the instance-wise consistency, under the circumstance of injecting backdoors into a trained clean model.

The main contribution of the paper includes two aspects:
(1) The paper observes an interesting phenomenon that the variations of parameters are always AWPs when tuning the trained clean model to inject backdoors, and provide theoretical analysis to explain this phenomenon.
(2) The paper first formulates the consistency of backdoored models, and proposes a novel anchoring loss to anchor or freeze the model behaviors on the clean data.


**Summary Of The Review:**

The paper focuses on a specific yet common circumstance, injecting backdoors into a trained clean model in backdoor learning, explains the phenomenon of AWP in this circumstance and provides interesting theoretical insights. The paper also models the concept of consistency in backdoor learning, and proposes a novel logit anchoring method for better instance-wise consistency. The extensive experiments and analysis are comprehensive and solid. However, the reasons to choose the metrics or indicators to evaluate the instance-wise consistency are not clear and need further clarification. In general, this paper is novel and solid, and I recommend a strong acceptance.

---

> ### Author Response · Authors · 2021-11-20
> **Responses to Reviewer rNcw**
>
> Thank you for your positive review. Here is our response.
>
> **Q1**. We choose these five metrics in order to evaluate the instance-wise consistency.
>
> **A1**. The Pearson correlation of the performance indicator (such as ACC, F-1, or BLEU) only considers the consistency of predicted labels. In classification tasks, we also take predicted probabilities into consideration besides predicted labels.
>
> For example, if model A predicts [0.9, 0.1] for the label 0 and 1, and model B predicts [0.6, 0.4] for the label 0 and 1, they are totally consistent in this instance in labels, but vary in probabilities or confidence. The KL-div and other metrics except correlation take the consistency in probabilities or confidence into consideration. However, in other tasks, such as text generation, the probabilities or confidence are not available, only Pearson correlation is available and we cannot calculate KL-div or other indicators directly. Therefore, we may need both KL-div and Pearson correlation.
>
> Due to the space limitation of the main manuscript, we will discuss it in the revised version (Appendix B.3).

---

> > ### Comment · Reviewer_rNcw · 2021-11-22
> > **Comments have been well addressed**
> >
> > I have checked the detailed response from the authors. It has fully addressed all my concerns. Overall, I think it is a very excellent work in NLP security domain. Solid experiments strongly support their claim and theoretical explanation are given. I believe this work will arouse a substantial impact for future work to follow. Therefore, I totally agree with the acceptance recommendation from all the other three reviewers, and would like to recommend a strong acceptance of this work for oral or spotlight.

---

### Decision · Program_Chairs · 2022-01-20

**Decision:**

Accept (Poster)

**Comment:**

This paper proposes measures of consistency between back-doored and clean models, proposes regularization using those consistency measures, and showcases that such trained models indeed exhibit better consistency. Also, it is demonstrated that the fine-tuned model does not deviate too far from the original clean model. The reviewers' comments are all well addressed. Some concerns related to the notion of consistency and how it relates to the detection of backdoors are still left open, but the reviewers seem to be satisfied with the answers. Given the overwhelmingly positive reviews, I propose accept.